# Dynamic allostery in the peptide/MHC complex enables TCR neoantigen selectivity

Jiaqi Ma [1,2,7], Cory M. Ayres [1,2,7], Chad A. Brambley [1,2], Smita S. Chandran[3,4], Tatiana J. Rosales[1,2], W. W. J. Gihan Perera[1,2], Bassant Eldaly [1,2], William T. Murray[3,4], Steven A. Corcelli[1], Evgenii L. Kovrigin[1], Christopher A. Klebanoff [3,4,5,6] & Brian M. Baker [1,2] ✉

The inherent antigen cross-reactivity of the T cell receptor (TCR) is balanced by high specificity. Surprisingly, TCR specificity often manifests in ways not easily interpreted from static structures. Here we show that TCR discrimination between an HLA-A*03:01 (HLA-A3)-restricted public neoantigen and its wild-type (WT) counterpart emerges from distinct motions within the HLA-A3 peptide binding groove that vary with the identity of the peptide's first primary anchor. These motions create a dynamic gate that, in the presence of the WT peptide, impedes a large conformational change required for TCR binding. The neoantigen is insusceptible to this limiting dynamic, and, with the gate open, upon TCR binding the central tryptophan can transit underneath the peptide backbone to the opposing side of the HLA-A3 peptide binding groove. Our findings thus reveal a novel mechanism driving TCR specificity for a cancer neoantigen that is rooted in the dynamic and allosteric nature of peptide/MHC-I binding grooves, with implications for resolving long-standing and often confounding questions about T cell specificity.

Using their T cell receptor (TCR), T cells orchestrate cellular immunity by recognizing antigenic peptides bound and presented by major histocompatibility complex (MHC) proteins. Cross-reactivity is a hallmark of TCRs, ensuring a TCR repertoire limited to millions of clonotypes can accommodate a vastly larger array of potential ligands[1,2]. Paradoxically, however, TCRs are also highly specific and can show surprising sensitivity to subtle modifications to antigenic peptides. Although specificity can often be interpreted in the context of static structural features, such as amino acid substitutions that alter hot spots in the TCR-peptide/MHC interface or alter the conformation of the peptide in the MHC binding groove[3], in many cases, TCR sensitivity to peptide modifications cannot be readily interpreted from static structures alone. This has particularly been true for peptides presented by class I MHC proteins, notably in instances where TCRs sense changes to peptide primary or secondary anchors[4,5]. In one recent

case, different TCRs specific for the gp100[209] shared tumor antigen presented by HLA-A2 differentially sensed subtle variations in peptide position 2 amino acids that are buried deep in the "B" pocket of the MHC binding groove. In this case, changes as simple as replacing a methionine with leucine yielded TCR-dependent changes in receptor binding affinities in the absence of any discernable structural changes in the peptide/MHC complex[6]. Notably, position 2-modified variants of the gp100[209] peptide have been widely studied as potential heteroclitic cancer vaccines but have failed to achieve significant clinical benefit[7], in part because the anchor modifications render the WT and modified peptides antigenically distinct[8]. Observations similar to those with the gp100[209] tumor antigen have also been made with modified variants of other tumor antigens[6,9–13].

In addition to heteroclitic immunogens, TCR specificity emerging from anchor modification in the absence of structural explanations has

[1]Department of Chemistry and Biochemistry, University of Notre Dame, Notre Dame, IN, USA. [2]Harper Cancer Research Institute, University of Notre Dame, Notre Dame, IN, USA. [3]Human Oncology and Pathogenesis Program, Memorial Sloan Kettering Cancer Center (MSKCC), New York, NY, USA. [4]Center for Cell Engineering, MSKCC, New York, NY, USA. [5]Weill Cornell Medical College, New York, NY, USA. [6]Parker Institute for Cancer Immunotherapy, New York, NY, USA. [7]These authors contributed equally: Jiaqi Ma, Cory M. Ayres. ✉e-mail: brian-baker@nd.edu

implications for the design of personalized neoantigen vaccines using in silico prediction tools. Prediction efforts for such neoantigens often compare the properties of the mutant epitope to its wild-type (WT) counterpart and, in many cases, emphasize mutations at primary anchor positions due to their effect on MHC binding. Notably, vaccine predictions using current algorithms are routinely characterized by high rates of both false positives and false negatives[14].

We recently described a panel of TCRs that recognize a public neoantigen resulting from a hotspot mutation in the phosphoinositide 3-kinase p110α catalytic subunit (PI3Kα, encoded by the gene *PIK3CA*) that is presented by HLA-A*03:01 (HLA-A3)[15]. The neoantigen results from the substitution of leucine for histidine at the first primary anchor (position 2; sequence A**H**HGGWTTK → A**L**HGGWTTK). The mutation enhances peptide binding to HLA-A3, and the neoantigen drives T cell destruction of mutant tumors while leaving tumor cells producing wild-type PI3Kα unharmed. Although the crystallographic structures of the neoantigen and WT peptide/HLA-A3 complexes were nearly indistinguishable, we were unable to detect recognition of the WT peptide/HLA-A3 complex[15]. While weak responses are expected due to the weaker binding of the WT peptide to HLA-A3, no responses were seen at high, supraphysiological concentrations, despite the WT peptide binding the class I molecule as strong as and even dissociating more slowly than other peptides for which T cell responses have been frequently detected[5,10,16–18]. Altogether, these results indicate a mechanism for TCR discrimination distinct from differences in peptide-MHC binding or static structural features.

Our work with the PI3Kα neoantigen provides an opportunity to study TCR specificity in the absence of structural differences in the context of a clinically relevant public neoantigen. We show that TCR discrimination between the PI3Kα neoantigen and WT peptide results from dynamic allostery, in particular how the identity of the position 2 anchor influences motions within the peptide/HLA-A3 complex that in turn either block (for the WT peptide) or facilitate (for the neoantigen) a dramatic peptide conformational change that is required for TCR binding. Beyond the relevance for immunotherapy targeting mutant PI3Kα, our results highlight the dynamic and allosteric nature of peptide/MHC complexes. Such dynamic properties are not always apparent from static structures but are likely a key feature of immune recognition, with significant implications for T cell specificity and the therapeutic design efforts that depend on it, including the identification and optimization of immunogenic cancer neoantigens and the TCRs that recognize them.

## Results

### TCRs distinguish between the PI3Kα neoantigen and WT peptide despite nearly identical static structures

Our previous work studying the immunogenicity of the PI3Kα neoantigen indicated that, although T cell responses could be observed against the neoantigen, multiple neoantigen-specific TCRs were unable to recognize the WT peptide/HLA-A3 complex[15]. Although less potency is expected from the WT peptide given its weaker binding to HLA-A3, in other instances, similarly weak binding peptides have still elicited quantifiable functional responses[16,19,20]. This is exemplified by studies with the nonameric MART-1 tumor antigen (AAGIGILTV), whose MHC binding affinity is nearly the same as the PI3Kα WT peptide[17]. Yet despite its weak binding, measurable responses to the native MART-1 nonamer have been described for numerous T cell clones[5,16]. Moreover, the half-life of the WT PI3Kα peptide/HLA-A3 complex is more than twice as long as the half-life of the MART-1 nonamer/HLA-A2 complex[10,15].

The inability to detect T cell responses against the PI3Kα WT peptide was even more curious given our structural work with the neoantigen and WT peptide/HLA-A3 complexes that showed nearly identical conformations for the peptides in the binding groove[15]. Although we observed a slight 1–2 Å variance in the central backbone

resulting from ϕ/ψ bond differences in the glycines at positions 4 and 5 and originally hypothesized this could underlie specificity, the differences in the conformations of the peptides in these static structures are within the error limits recently established by analyses of structures of replicate class I peptide/MHC complexes[21]. Indeed, a second structure of the neoantigen/HLA-A3 complex lacked the variance in the peptide backbone (Fig. 1a; Supplementary Table 1). As the neoantigen undergoes a large conformational change in the center of the peptide when recognized by two unrelated neoantigen/HLA-A3-specific TCRs (termed TCR3 and TCR4)[15], we formulated a refined hypothesis that TCR specificity for the neoantigen emerges from differential motions within the binding groove.

We thus sought to resolve the mechanism basis for how PI3Kα neoantigen-specific TCRs can achieve selectivity for the mutant over the WT peptide. To confirm the inability of T cells to recognize the WT peptide, we first examined stimulation of T cells transduced with TCR4, among the most sensitive of our previously described TCRs. Co-culture experiments with HLA-A3+ antigen-presenting cells measuring the degranulation marker CD107a showed a clear dose-response curve for the neoantigen, with an $EC_{50}$ value near 100 nM (Fig. 1b). However, no stimulation was evident with the WT peptide, even at peptide concentrations as high as 10 μM.

The simplest explanation for the functional data is that, despite the overlapping static peptide/HLA-A3 structures, TCR binding to the WT peptide/HLA-A3 complex is weaker than TCR binding to the neoantigen complex. We thus assessed TCR discrimination between the neoantigen and WT peptide in a direct binding experiment that controlled for the weaker binding of the WT peptide to HLA-A3. We coupled TCR4 to one flow cell of a surface plasmon resonance (SPR) sensor surface. In an adjacent flow cell, we coupled the recently described single-chain TCR variant (scTv) s3-4. As s3-4 binds class I MHC proteins away from the peptide binding groove with an affinity independent of the bound peptide[22], this reagent allowed us to verify the integrity of the WT peptide/HLA-A3 complex during the experiment. To help promote sample stability, we performed the experiments at the reduced temperature of 4 °C.

As before, TCR binding to the neoantigen complex was readily detected, with the experiments yielding a $K_D$ of $62 \pm 6$ μM for TCR4 binding the neoantigen/HLA-A3 complex (Fig. 1c). Consistent with the functional experiment, no binding was detectable to the WT complex. However, strong binding to the s3-4 scTv positive control was apparent with both peptide/HLA-A3 samples, with a measured $K_D$ of $1.7 \pm 0.2$ μM for the neoantigen complex and an identical $K_D$ of $1.8 \pm 0.1$ μM for the WT complex (Fig. 1d). Thus, our inability to detect TCR binding to the WT peptide/HLA-A3 complex is attributable to the ability of the TCR to distinguish between the neoantigen and the WT peptide.

### TCR recognition of the neoantigen is critically dependent on the rotation of the central tryptophan residue

The PI3Kα neoantigen is notable in that, upon recognition by two unrelated TCRs, the peptide undergoes a large conformational change. In the free (i.e., not TCR-bound) peptide/HLA-A3 complex, the side chain of the tryptophan at position 6 (pTrp6) packs against the HLA-A3 α1 helix, whereas in the structures with two different neoantigen-specific TCRs (TCR3 and TCR4), the side chain and the neighboring backbone units have flipped around the axis of the peptide backbone, with the pTrp6 side chain buried within the groove and now packed against the α2 helix (Fig. 2a; Supplementary Fig. 1). When all atoms of the peptide are considered, the conformational change (which we refer to as a 'flip' for pTrp6) is among the largest yet observed for TCRs binding nonameric peptides presented by class I MHC proteins (Fig. 2b).

While the flip of pTrp6 is a distinctive structural feature of PI3Kα neoantigen recognition by TCR3 and TCR4, its overall importance in TCR recognition is unclear, as the flipped side chain is not contacted in the complexes with either TCR3 or TCR4. In the structure with TCR4,

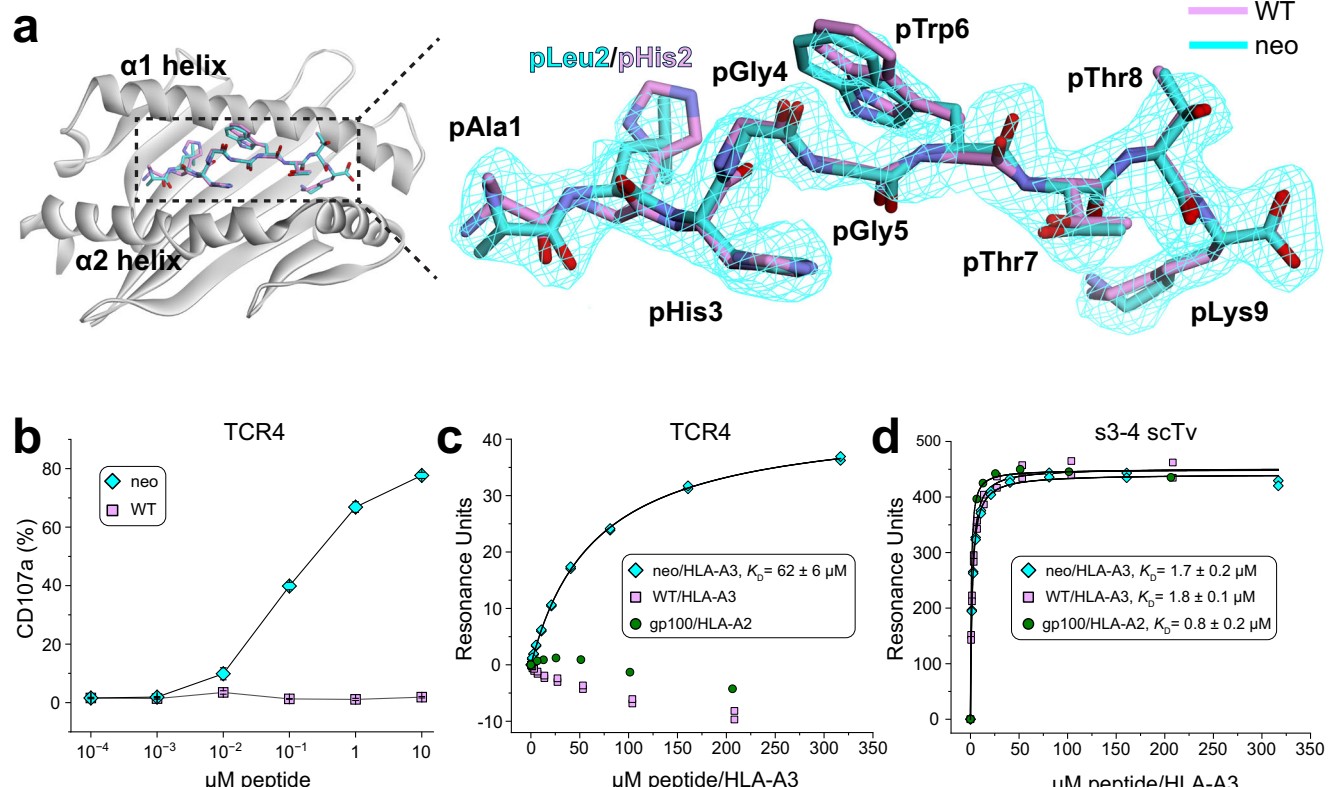

**Fig. 1 | TCRs distinguish between the PI3Kα neoantigen and WT peptide presented by HLA-A3 despite overlapping structures. a** Comparison of the WT peptide and a second crystallographic structure of the PI3Kα neoantigen in the HLA-A3 binding groove, demonstrating the closely overlapping structures. The neoantigen is cyan, the WT peptide is pink; this color scheme is maintained throughout all figures. Electron density of the neoantigen is from a $2F_o$-$F_c$ composite OMIT map calculated with simulated annealing, contoured at $1\sigma$. Superimposition is by the $C\alpha$ atoms of residues 1-180 of the HLA-A3 peptide binding grooves, yielding an all common atom RMSD for the peptides of 0.8 Å. **b** Measurement of T cell function via the degranulation marker CD107a. T cells expressing the neoantigen-specific receptor TCR4 were co-cultured with HLA-A3$^+$ antigen presenting cells in the presence of increasing concentrations of neoantigen or WT peptide. Although neoantigen recognition is clear, there is no recognition of the WT peptide. Data are absolute frequencies; points and error bars are means and standard deviations from three replicates. **c** SPR experiments of TCR4 with the neoantigen or WT peptide/HLA-A3 complexes. Although neoantigen recognition was quantifiable, no binding was detected with the WT peptide. gp100/HLA-A2 is an irrelevant negative control complex of the peptide IMDQVPFSV presented by HLA-A2, for which no binding was also detected. **d** The neoantigen and WT peptide/HLA-A3 complexes bind the s3-4 scTv with identical affinities, indicating the WT sample is stable over the course of an SPR experiment, and our inability to detect TCR binding to the WT peptide/HLA-A3 in panel (**c**) is due to TCR discrimination. The gp100/HLA-A2 control was also wellrecognized. $K_D$ and error values in panels (**c**) and (**d**) are the average and standard deviation of six and three replicates, respectively; titrations in both panels were performed at 4 °C for increased peptide/MHC stability.

the indole nitrogen of the side chain forms a hydrogen bond with the aromatic ring of tryptophan 147 of HLA-A3 (Fig. 2c). With TCR3, the slight change in the position of the side chain in the complex distorts the geometry of this interaction, although the side chain is still embedded in a complex electrostatic environment with structural water linking the indole nitrogen to the HLA-A3 $\alpha1$ helix (Fig. 2d). Substitution of pTrp6 in the neoantigen with alanine or glycine eliminates functional recognition with both TCRs[15]; however, neither of these substitutions probes the peptide conformational change.

We thus designed an experiment to directly probe the importance of the pTrp6 flip in the neoantigen. The tryptophan analog 3-benzothienyl-L-alanine (Bta) is isomorphic with tryptophan but replaces the indole nitrogen with a sulfur atom incapable of serving as a hydrogen bond donor (Fig. 3a). We reasoned that removing the hydrogen bond donor would destabilize the TCR-bound flipped peptide conformation, with a destabilized conformation resulting in weaker TCR binding. At the same time, removing the hydrogen bond donor by substituting pTrp6 with Bta should have little to no impact on the TCR-free peptide conformation, as in the neoantigen/HLA-A3 structure, the side chain at position 6 remains accessible to solvent (Fig. 3b). We verified that substitution of pTrp6 with Bta did not impact the binding of the peptide to HLA-A3 using differential scanning

fluorimetry: as expected, there was no impact on the stability of the peptide/HLA-A3 complex (Fig. 3c). We next determined the crystallographic structure of HLA-A3 presenting the Bta-substituted peptide (Supplementary Table 1). The structure showed there were no significant changes in peptide conformation, with the Bta6 side chain aligned against the HLA-A3 $\alpha1$ helix and the sulfur atom solvent exposed, mimicking the conformation of the neoantigen (Fig. 3d).

We assessed the recognition of the Bta-substituted neoantigen biochemically and functionally. Using SPR we could not detect binding of TCR4 or TCR3 to the complex of HLA-A3 with the Bta-substituted neoantigen (Fig. 3e), although this complex was readily recognized by the s3-4 scTv positive control (Fig. 3f). Functionally, the Bta-substituted neoantigen also failed to drive cytokine secretion by T cells transduced with TCR3 or TCR4 when co-cultured with HLA-A3$^+$ antigen presenting cells, although the unmodified neoantigen control was well-recognized (Fig. 3g). TCR recognition of the PI3Kα neoantigen is thus critically dependent on the flip of pTrp6.

## Conformational sampling of pTrp6 differs between the neoantigen and WT peptide

Conformational changes that occur upon binding are often reflected in the motions of the unbound protein[23]. To investigate these, and to

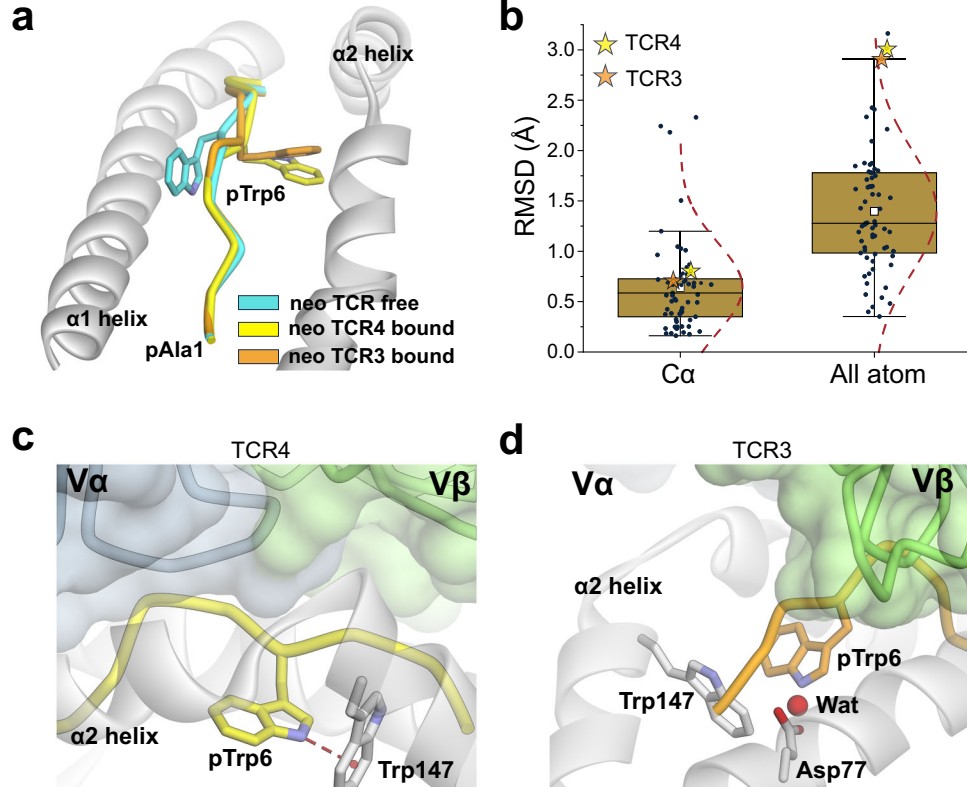

**Fig. 2 | TCR binding to the neoantigen induces a large conformational change in the peptide that leads to new peptide-HLA interactions. a** Illustration of the peptide conformational change that occurs upon the binding of TCR3 and TCR4 to the neoantigen/HLA-A3 complex. The tryptophan at position 6 has flipped from aligning against the α1 helix to nestling between the peptide backbone and the α2 helix (see also Supplementary Fig. 1). **b** Distribution of conformational changes that occur in nonameric class I MHC complexes upon TCR binding as measured by bound vs. free RMSDs for Cα atoms and all peptide atoms. The RMSD values for the neoantigen upon recognition by TCR3 or TCR4 are shown as orange and yellow stars, respectively. The white squares in the box plots give the average, the gold boxes the interquartile ranges (IQR), lines in the gold boxes the medians, and the

whiskers 1.5 × IQR. Although the changes for the backbone are only slightly above the mean, when considering all peptide atoms, the conformational change in the neoantigen is among the largest seen upon TCR recognition of nonamers. Data are from structural analysis of available PDB structures as described in the Methods. **c** In the TCR4 complex, the pTrp6 side chain is not contacted by the TCR. The indole nitrogen of the flipped conformation of pTrp6 forms a NH-π hydrogen bond with Trp147 of the HLA-A3 α2 helix. **d** In the complex with TCR3, the pTrp6 side chain is also not contacted by the TCR. The slight repositioning of the pTrp6 side chain in the complex with TCR3 distorts the interaction of pTrp6 with Trp147, although a structural water in close proximity bridges pTrp6 to Asp77 of the α1 helix.

assess any differences in the dynamics of the PI3Kα neoantigen and WT peptide bound to HLA-A3, we studied the motions of the peptides with a series of computational simulations. We first performed fully atomistic, unrestrained molecular dynamics (MD) on the peptide/HLA-A3 complexes, simulating each complex for 2 μs in explicit solvent. We examined peptide motion by computing mass-weighted root mean square fluctuations (mwRMSFs) for each amino acid of the two peptides. The neoantigen showed high fluctuations in its central bulge (positions 4 to 7) but was more rigid at both termini (Fig. 4a). The WT peptide showed similar high fluctuations in the center but also had high fluctuations in its N-terminal region, consistent with its suboptimal position 2 anchor of histidine.

As high fluctuations were seen in the centers of both peptides, we asked if either the neoantigen or WT peptide sampled the flipped conformation. For both sets of trajectories, we computed the RMSD of pTrp6 from its conformation in the TCR4 ternary complex, selecting this metric due to TCR4's higher affinity and potency compared to TCR3. The RMSD analysis indicated that, despite the high fluctuations, neither peptide sampled the TCR-bound state during the simulations (Fig. 4b). However, this analysis also suggested that conformational sampling differed between the two peptides. To further investigate, we analyzed the peptides during the two sets of trajectories using a metric referred to as the D-score, which was developed to evaluate conformational properties of antibody loops and used recently to describe

the conformations of peptides in class I MHC binding grooves[24,25]. The D-score provides a convenient way to compare differences in backbone torsion angles and ranges from 0 when two amino acids have identical ϕ/ψ bond angles to 8 when both angles differ by 180°. When the neoantigen and WT peptide were compared this way, the average D-score was low at the termini but peaked in regions from positions 4 to 7 (Fig. 4c). The divergent behavior of the neoantigen and WT peptides remained consistent across three additional independent pairs of simulations (Supplementary Fig. 2). Thus, although the neoantigen and WT peptide show high fluctuations across their central regions, the conformations they sample differ.

To visualize the space sampled by the centers of the neoantigen and WT peptides, we defined 3D grids centered around pTrp6 of the neoantigen and WT peptide, with a spacing of 0.1 Å. We then tabulated the occupancy of each resulting voxel by the atoms of pTrp6. As expected from the fluctuation data, we observed dispersed sampling for both the neoantigen and WT peptides (Fig. 4d, e). Consistent with the D-score analysis, sampling differed between the two peptides. The tryptophan side chain in the WT peptide remained close to its crystallographic conformation but also sampled states above the binding groove that left the side chain solvent exposed (Fig. 4d). In contrast, the side chain in the neoantigen sampled states more recessed in the binding groove, including states which placed the side chain underneath the peptide bulge (Fig. 4e).

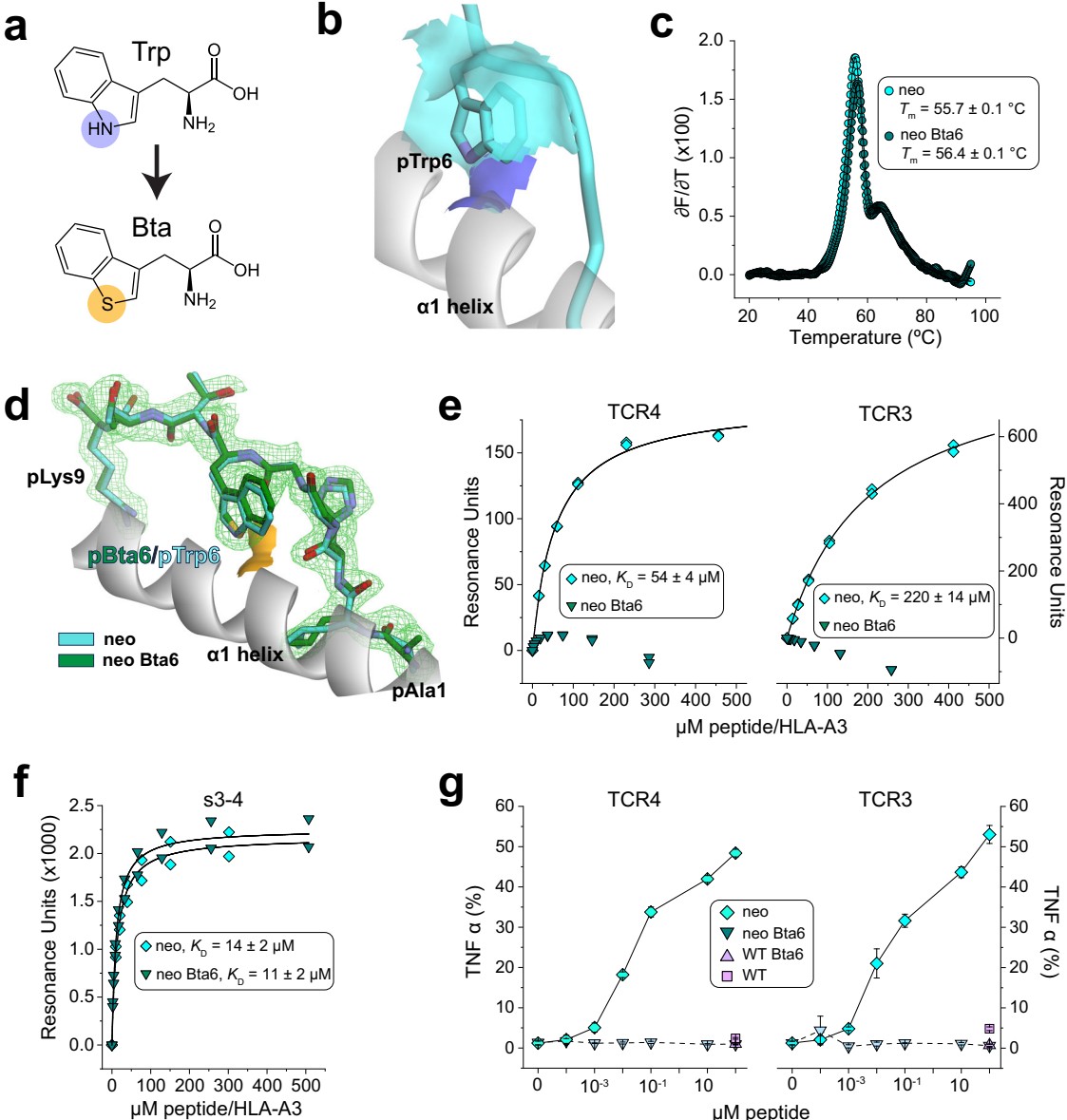

**Fig. 3 | TCR recognition of the neoantigen presented by HLA-A3 is critically dependent on the flip of the tryptophan at position 6. a** The tryptophan analog Bta replaces the indole NH with a sulfur atom, eliminating the capacity of the tryptophan side chain to serve as a hydrogen bond donor. **b** In the structure of the free neoantigen/HLA-A3 complex, the pTrp6 side chain remains accessible to solvent. The solvent accessible surface of the indole nitrogen is blue; the surface of the carbon atoms is cyan. **c** Substitution of pTrp6 with Bta does not alter peptide binding to HLA-A3 as indicated by differential scanning fluorimetry. Datapoints indicate the temperature derivative of the fluorescence ratio; only every 5th data-point is shown for clarity. $T_m$ and error values are the average and standard deviation of four replicates. **d** Substitution of pTrp6 with Bta does not alter the structural properties of the neoantigen in the HLA-A3 binding grooves as shown by the crystallographic structure of the Bta6-neoantigen/HLA-A3 complex. The Bta-substituted peptide is superimposed on the unsubstituted neoantigen from the replicate structure determined here. Superimposition is via the Cα atoms of residues 1-180 of the HLA-A3 peptide binding groove, yielding an all common atom RMSD for the peptides of 0.8 Å. The orange surface shows the solvent accessibility of the Bta sulfur atom to compare with that of the indole nitrogen in panel (**b**).

Electron density of the Bta-substituted neoantigen is from a $2F_o$-$F_c$ composite OMIT map calculated with simulated annealing, contoured at 1σ. **e** SPR experiments show no detectable binding of TCR4 or TCR3 to the Bta-substituted neoantigen/HLA-A3 complex, although binding to the non-substituted neoantigen complex was quantifiable. Experiments were performed at 25 °C; $K_D$ and error values are the average and standard deviation of three replicates. **f** Control SPR experiments confirming binding of the Bta-substituted and non-substituted neoantigen/HLA-A3 complexes to the peptide-independent s3-4 scTv. Experiments were performed at 25 °C; $K_D$ and error values are the average and standard deviation of four replicates. **g** Measurement of T cell function via production of the cytokine TNFα. T cells expressing either TCR4 or TCR3 were co-cultured with HLA-A3+ antigen presenting cells in the presence of increasing concentrations of Bta6-substituted or unmodified neoantigen. Although unmodified neoantigen recognition is clear, there is no recognition of the Bta6-modified version. The unmodified and Bta6-modified WT peptides at the highest concentration were included as controls. Data are absolute frequencies; points and error bars represent means and standard deviations from three replicates.

To examine individual conformations adopted, we calculated pairwise RMSDs of pTrp6 in the two simulations following superimposition of the peptide binding grooves (Supplementary Fig. 3A). The results clustered into eight distinct conformations (Supplementary Fig. 3B, C), with the WT peptide sampling six of those conformations and the neoantigen sampling two. Only one conformation (cluster 3) was sampled by both peptides, and the coordinates of the TCR4-bound neoantigen (cluster 7) were not sampled at

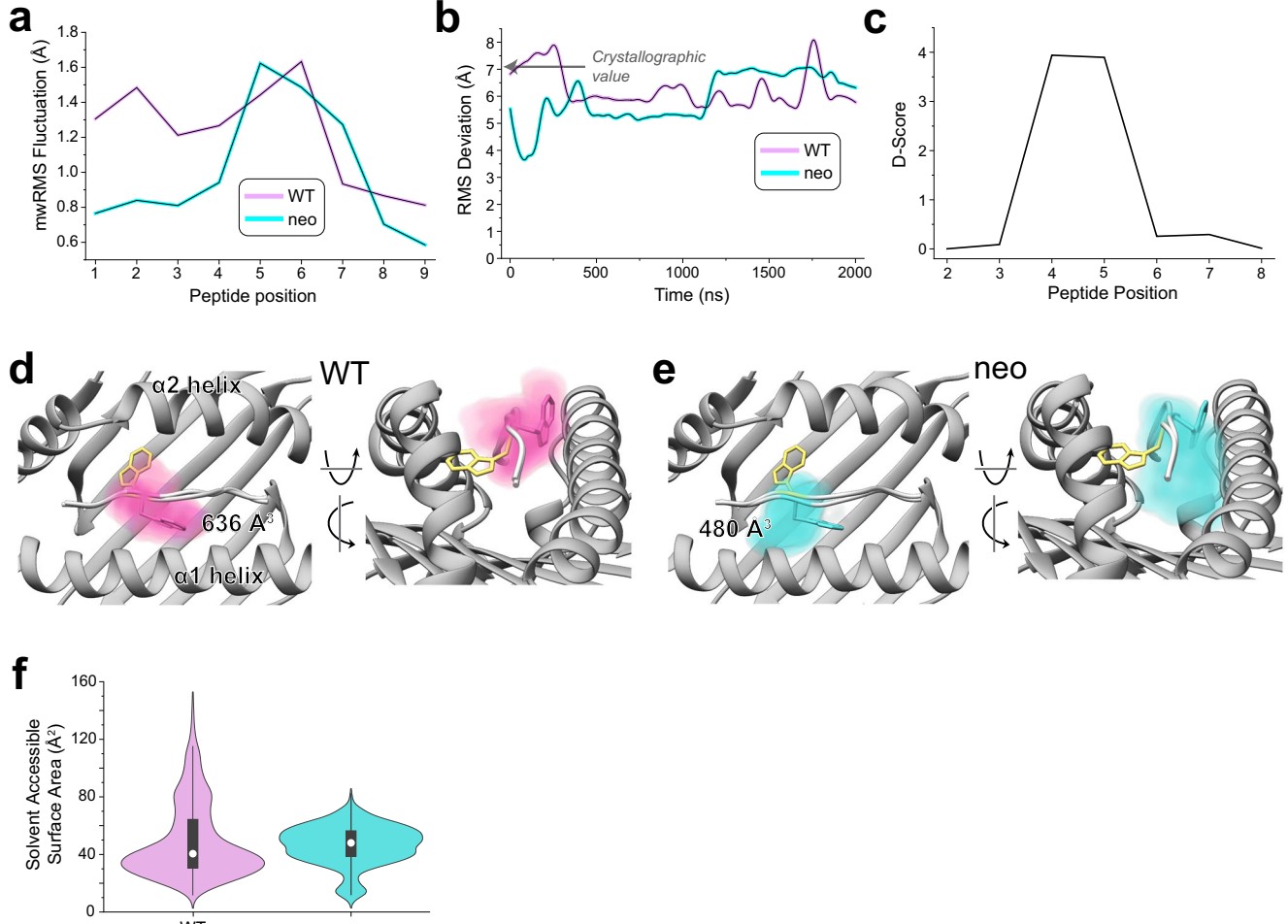

**Fig. 4 | Conformational sampling differs in the neoantigen and WT peptide/HLA-A3 complexes. a** Mass-weighted RMS fluctuations for each amino acid of the neoantigen and WT peptide in the HLA-A3 binding groove from 2 μs of unrestrained, fully atomistic molecular dynamics simulations. The central regions of both peptides are mobile, as is the N-terminal half of the WT peptide but not the neoantigen, consistent with the neoantigen's more optimal position 2 anchor. **b** Despite high mobility, neither peptide samples the TCR-bound conformation, as shown by the RMSD of the pTrp6 amino acid relative to its position in the ternary complex with TCR4 during the simulations (data were smoothed using LOWESS; see Supplementary Fig. 13 for unsmoothed data). The value of 7.1 Å from super-imposition of the TCR4-bound and free structures is indicated by the arrow. **c** Conformational sampling differs across the centers of the peptide backbones of the neoantigen and WT peptide as indicated by a D-score analysis comparing φ/ψ

bond angles derived from the average peptide conformations during the two simulations. **d** Conformational space occupied by the pTrp6 side chain during the simulation with the WT peptide. Color density reflects degree of sampling (voxels sampled <10% of the time are excluded), values give volumes of sampled space. Note the tendency for the pTrp6 side chain to move above the peptide backbone: although it reaches over, it does not flip to the TCR-bound state as indicated by panel (**b**). **e** As in panel (**d**), except volume occupied by pTrp6 during the simulation with the neoantigen. Note the tendency for the pTrp6 side chain to move under the peptide backbone. **f** Solvent accessible surface areas of the pTrp6 side chain during the WT and neoantigen simulations. Although the average values are similar, the WT peptide samples a much wider range of values, reflecting the volumetric analyses in panels (**d**) and (**e**). White circles in the violin plots show averages, black rectangles the IQR, and vertical lines 1.5 × IQR.

all. In agreement with the conformational clustering analysis, the greater conformational variability for the pTrp6 side chain in the WT peptide was further reflected in an analysis of the side chain's solvent-accessible surface area during the two simulations, which showed similar average levels of accessibility, but a much wider range for the WT peptide vs. the neoantigen (Fig. 4f).

**Different conformational sampling of pTrp6 is linked with differential motions of position 2**

The differential dynamics in the N-terminal halves of the neoantigen and WT peptide revealed by the RMSF data (Fig. 4a) prompted us to examine conformations adopted by peptide position 2. We thus performed the same visualization analysis for position 2 that we performed for pTrp6. Consistent with the RMSF data, the suboptimal position 2 histidine of the WT peptide (pHis2) was highly mobile, sampling a large volume within the HLA-A3 B pocket, whereas the

position 2 leucine of the neoantigen (pLeu2) was tightly constrained (Fig. 5a). To examine the conformations sampled by these amino acids, we performed pairwise RMSD and clustering analyses as on pTrp6, but now on position 2. The more dynamic pHis2 of the WT peptide clustered into six conformations, compared to just two for the rigid pLeu2 of the neoantigen (Supplementary Fig. 4A, B). The predominant cluster of pLeu2 of the neoantigen was essentially identical to that of the crystallographically observed state. In contrast, the clusters for the more mobile pHis2 of the WT peptide had various alterations of the side chain's χ1 and χ2 angles.

Examining the pHis2 clusters of the WT peptide in more detail, we observed that one of the clusters with an alternate pHis2 position showed a large rotation in the χ1 torsion angle of the neighboring Tyr99 in the floor of the HLA-A3 binding groove, with the Tyr99 side chain rotating ~140° to avoid a steric clash with the histidine (cluster 1, with a population of 23%). This rotation placed Tyr99 directly under

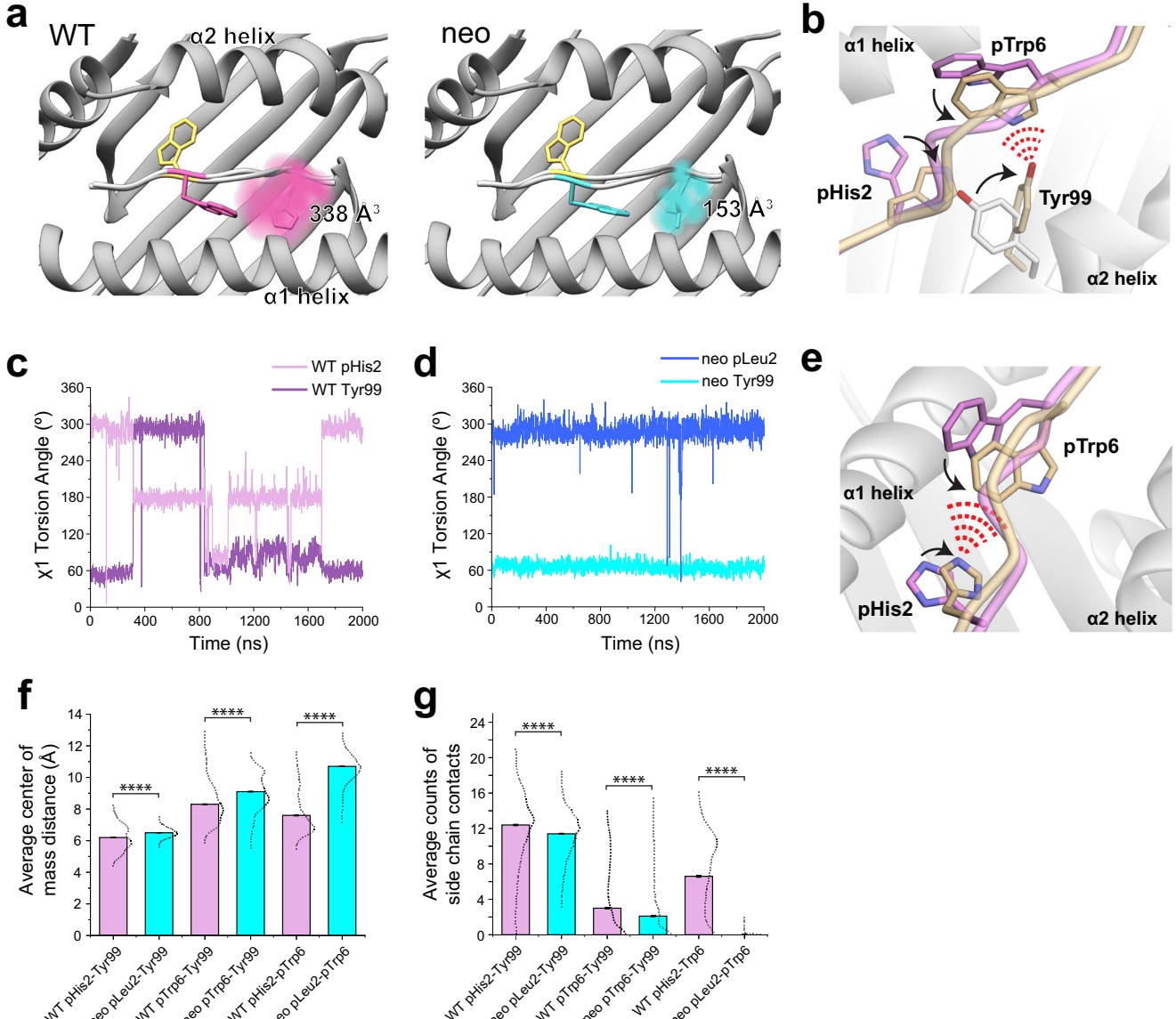

**Fig. 5 | The extensive motions of pHis2 in the WT peptide/HLA-A3 complex lead to more interatomic interactions with pTrp6 as it moves in the binding groove.** **a** Volume occupied by pHis2 during the simulations with the WT peptide (left) or pLeu2 during the simulations with the neoantigen (right). Color density reflects degree of sampling (voxels sampled <10% of the time are excluded), values give volumes of sampled space. Note the greater mobility and sampled volume of pHis2 in the WT peptide. The TCR4-bound conformation of pTrp6 is colored yellow. **b** In the simulation with the WT peptide, the movement of pHis2 is associated with an alternate conformation for Tyr99 of HLA-A3 and formation of contacts (red dashes) between the altered Tyr99 conformer and pTrp6 in the WT peptide. Structural snapshot is representative of cluster 1 in Supplementary Fig. 4B. **c** The χ1 torsion angles of Tyr99 of HLA-A3 and pHis2 of the peptide during the WT peptide/HLA-A3 simulation. During the first half of the simulation, the rotation in pHis2 induces a rotation in Tyr99 (in the latter half of the simulation, the peptide N-terminus has become less recessed in the binding groove, decoupling pHis2/Tyr99 motion).

**d** The χ1 torsion angles of pLeu2 and Tyr99 remain fixed in the simulation with the neoantigen. **e** Other conformations of pHis2 in the simulation with the WT peptide show contacts (red dashes) between the side chains of the histidine and pTrp6. Structural snapshot is representative of cluster 2 in Supplementary Fig. 4B. **f** Average distance between the centers of mass of the side chains of the position 2 amino acid, pTrp6, and Tyr99 during the simulations with the neoantigen and WT peptide. Average distances are all closer in the WT simulation. Error bars are SEM, calculated from the 2000 1 ns frames of the 2 µs simulations. Dotted lines indicate the distribution of distances calculated from the 2000 datapoints. **** = p < 0.0001. **g** Average counts of side chain contacts between the side chains of the position 2 amino acid, pTrp6, and Tyr99 during the simulations with the neoantigen and WT peptide. Contacts are all higher in the WT simulation. Error bars are SEM, calculated from the 2000 1 ns frames of the 2 µs simulations. Dotted lines indicate the distribution of distances calculated from the 2000 datapoints. **** = p < 0.0001. p values in panels (**f**) and (**g**) are from an unpaired t test.

the bulge of the peptide backbone and adjacent to pTrp6, which had entered the base of the groove (Fig. 5b). Interrogating the WT simulation revealed a connection between the rotation of χ1 of Tyr99 and pHis2 (Fig. 5c). In contrast, in the simulation with the neoantigen, the χ1 torsion angles of pLeu2 and Tyr99 did not deviate from their initial positions (Fig. 5d). Visualization of the space sampled by Tyr99 as performed for pTrp6 and pHis2/pLeu2 confirmed it was highly mobile

in the WT simulation, but rigid in the neoantigen simulation (Supplementary Fig. 5A).

We examined the other conformational clusters for pHis2 in the WT simulation and found that a majority (total population of simulation time of 63%) placed the histidine adjacent to pTrp6 as it entered the base of the groove (Fig. 5e). Together with the Tyr99 data, this prompted us to examine the distances between the position 2 side

chain and the side chains of pTrp6 and Tyr99 in the two simulations. The comparison confirmed that not only were pHis2 and Tyr99 more dynamic in the WT simulation, but compared to the neoantigen simulation, they were on average closer to pTrp6 (Fig. 5f; Supplementary Fig. 5B). The closer proximity in the WT simulation led to the formation of substantial contacts between pHis2, pTrp6, and Tyr99 (Fig. 5g; Supplementary Fig. 5C). However, such inter-residue interactions were reduced or, for the pLeu2-pTrp6 contacts, essentially absent in the neoantigen simulation. The overall picture is that, for the WT peptide, the greater motion of the pHis2 side chain leads to numerous direct and indirect inter-residue interactions with pTrp6 as it enters the base of the binding groove. However, these interactions are far less significant for the neoantigen, permitting pTrp6 to reach further into the base of the HLA-A3 peptide binding groove as shown in Fig. 4e.

We next asked why the pHis2 side chain in the wild-type peptide was so mobile in the HLA-A3 B pocket compared to the leucine of the neoantigen. HLA-A3 shows a strong preference for hydrophobic amino acids in its hydrophobic B pocket[26,27]. The deeper part of the pocket, into which pLeu2 of the neoantigen extends, is notably preceded by a hydrophobic depression. In the structure with the wild-type peptide, pHis2 lies in this depression (Supplementary Fig. 5D). From the structure of the wild-type peptide/HLA-A3 complex, we computed the $pK_a$ of pHis2 of the wild-type peptide using continuum electrostatics[28]. This resulted in an estimated $pK_a$ of 4.9, which would leave the histidine uncharged at physiological pH. The computed $pK_a$ for pHis2 in the free peptide was 6.1, indicating an electrostatic destabilization of 1.6 kcal/mol. The compression of the energy differences between conformational states resulting from this destabilization translates into a flatter energy landscape and thus more rapid conformational interconversion, as has been seen in other cases when polar or charged amino acids are buried in hydrophobic environments[29].

## The flip in the neoantigen occurs with a high energy peptide-limbo mechanism

Our inability to observe the flip of the pTrp6 side chain in the neoantigen during traditional MD simulations suggests a high energy barrier. To test this, we measured the association rate ($k_{on}$) for binding of TCR4 to the neoantigen/HLA-A3 complex. We performed this by first measuring the dissociation rate ($k_{off}$) for TCR4. Using the same SPR configuration we used to measure binding affinity in Fig. 1c, we determined a $k_{off}$ of $0.039 \pm 0.006 \, s^{-1}$ (Supplementary Fig. 6A). From the relationship $k_{on} = k_{off}/K_D$, this yielded a $k_{on}$ of $629 \pm 14 \, M^{-1} s^{-1}$. Compared to other TCRs, this is an exceptionally slow association rate[30]. For example, under the same conditions, the archetypical anti-viral TCR A6 binds the HTLV-1 Tax peptide presented by HLA-A2 ~20-fold faster, with a $k_{on}$ of $1.1 \times 10^4 \, M^{-1} s^{-1}$[31]. Notably, the binding of A6 to Tax/HLA-A2 also occurs with a conformational change in the peptide backbone as well as conformational changes in both TCR CDR3 loops[32–34]. Repeating the experiment but with the side-binding scTv s3-4, we measured a 25-fold faster $k_{on}$ of $1.5 \times 10^4 \, M^{-1} s^{-1}$. Although multiple factors influence rates of protein association, the rates of conformational changes are a major contributor[35,36]. The very slow $k_{on}$ for the binding of TCR4 is consistent with a high barrier and thus slow rate for the flip of pTrp6 in the neoantigen.

To better study how the peptide crosses this barrier and moves to the TCR-bound state, we used weighted ensemble molecular dynamics simulations (WEMD) to identify potential motional pathways. Rather than sampling stable states and waiting for low-probability transitions, WEMD uses multiple simulations run in parallel and tracks the progression of each toward a defined target state. To overcome high barriers, WEMD reweights trajectories based on their progress toward the target state, thereby sampling rare events without introducing biases such as steering forces[37,38]. We performed WEMD beginning with the neoantigen/HLA-A3 complex either in its TCR-free

conformation (a forward simulation) or in its conformation in the ternary complex with TCR4 (a reverse simulation, performed with the TCR removed). Target states were defined as the crystallographic structures of the TCR4-neoantigen/HLA-A3 ternary complex (for the forward simulation) or the TCR-free neoantigen/HLA-A3 complex (for the reverse simulation). The progress of the simulation was tracked by the RMSD of pTrp6 between the trajectory and the target state. An RMSD < 1 Å was defined as a successful transition.

While the forward simulation approached but did not fully flip even after 1300 iterations, we found 109 trajectories containing successful transitions over 970 iterations in the reverse simulation (Supplementary Fig. 6B, C). These results are consistent with a high energy barrier in the forward direction and a lower barrier in reverse (Fig. 6a). Remarkably, each of the successful trajectories from the reverse simulation showed the pTrp6 side chain moving underneath the peptide backbone, evoking a limbo dance (Fig. 6b; Supplementary Movie 1). This was evident not only from visual inspection, but also quantitatively from an analysis of the solvent accessible surface area of pTrp6 during the transitions, which increased as the peptide moved from its TCR-bound to TCR-free conformation but did not approach the value of an exposed amino acid tryptophan oriented above the peptide bulge (Fig. 6c). Additionally, even though we did not observe a complete transition in the forward simulations, the trajectory that most closely approached the target placed the pTrp6 side chain underneath and on the other side of the peptide backbone near to its position in the TCR-bound state (Supplementary Fig. 6D). While a "peptide limbo" transition was initially surprising, the pathway revealed evokes the conformations seen in the traditional MD analysis of the neoantigen, which as noted above did not flip but did place the pTrp6 side chain near the base of the groove (Fig. 4e).

The under-peptide transition suggests the presence of empty space between the peptide backbone and the floor of the HLA-A3 protein. To examine this, we computed open volumes (or cavities) in the neoantigen/HLA-A3 complex. This analysis revealed a large cavity between the peptide and the HLA-A3 binding groove floor. Cavity volumes in proteins are sensitive to atomic positions, and the cavity volume was 280 Å³ or 400 Å³ depending on which neoantigen/HLA-A3 coordinates were used (Supplementary Fig. 7A, B). In either case though, as the molecular volume of tryptophan is ~160 Å³, the space needed to accommodate the motional pathway indicated by WEMD is clearly available. We examined other peptide/MHC complexes in the HLA-A3 superfamily whose structures were available and determined that cavities between the peptide backbone and floor of the groove are common, as analysis of 40 different structures revealed an average cavity size of 174 Å³ with a large standard deviation of 112 Å³ (Supplementary Table 2). We also surveyed other structures of nonameric peptides bound to HLA-A proteins and found several examples of peptides with large position 6 side chains positioned under the peptide backbone. The structure of the peptide IIGWMWIPV bound to HLA-A2 is a notable example: the conformation of the peptide backbone is nearly identical to that of the neoantigen, yet the tryptophan at position 6 is oriented down towards the base of the groove[39], resembling an intermediate for the under-peptide transition of the PI3Kα neoantigen (Supplementary Fig. 7C). The architecture of the HLA-A3 peptide binding groove is thus clearly compatible with an under-peptide, limbo dance pathway.

To check the validity of the under-peptide pathway, we performed steered molecular dynamics (SMD) to rotate the pTrp6 side chain under the backbone. SMD differs from WEMD in that, rather than searching for a pathway of motion, a pathway is stipulated through applied force. We used an approach termed enforced rotation, in which a set of atoms (defined as the rotation group) are pulled around a user-defined axis at a constant angular velocity[40]. Beginning with the structure of the neoantigen bound to HLA-A3, we assigned all pTrp6

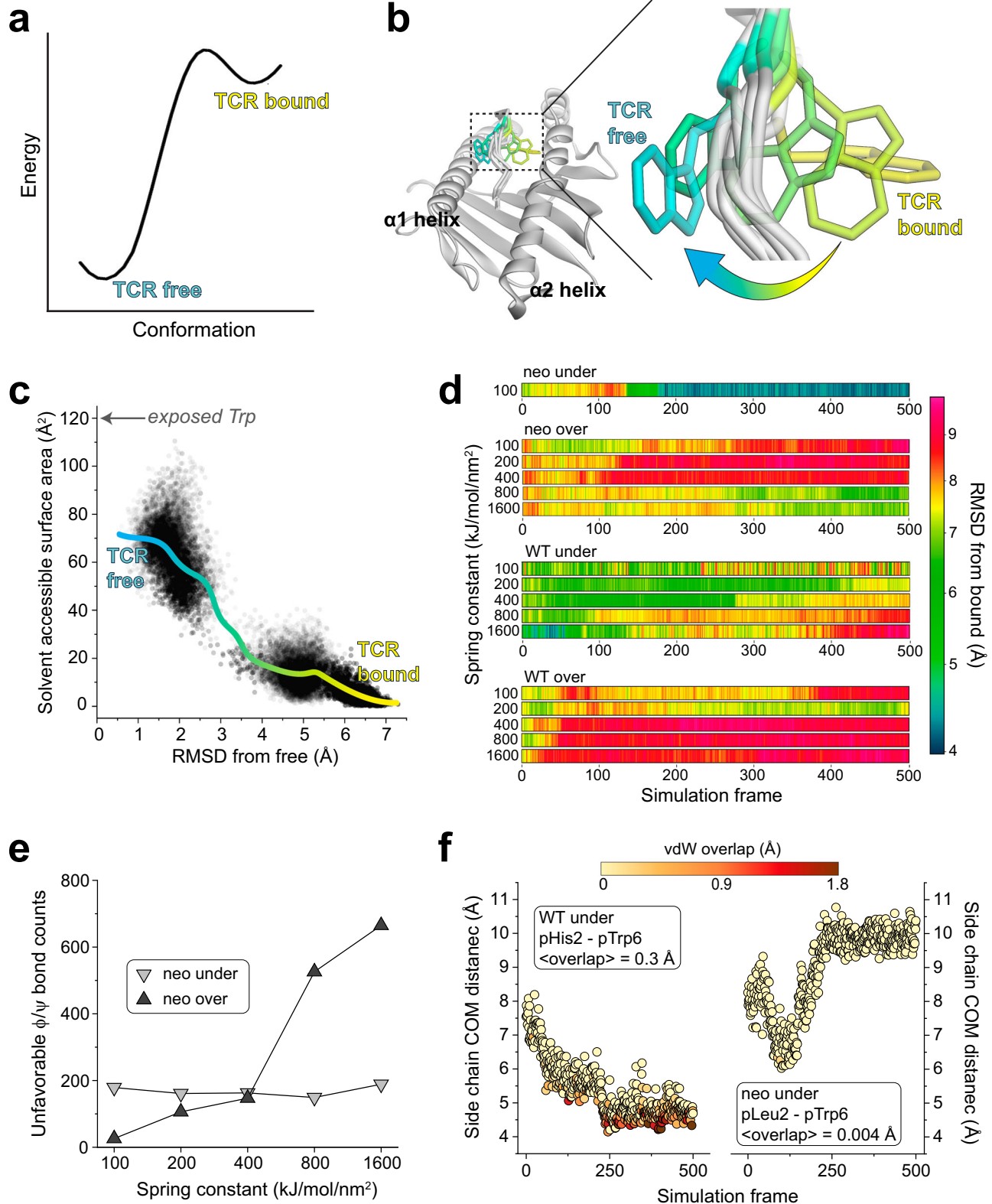

atoms to the rotation group and defined a rotational axis as a vector through the peptide backbone (Supplementary Fig. 8A). With an initial spring constant of 100 kJ/mol/nm², we first rotated pTrp6 in the neoantigen complex underneath the peptide backbone. We readily observed under-peptide rotation, with the pTrp6 side chain adopting a conformation near that seen in the structure with TCR4 within 500 ps of enforced rotation (Fig. 6d, neo under). Subsequent traditional MD initiated from this state resulted in the side chain equilibrating rapidly

into a conformation nearly identical to that in the TCR4 ternary complex (Supplementary Fig. 8B).

As an alternative to rotating the pTrp6 side chain under the peptide backbone, we inverted the directionality of the rotational axis to rotate the side chain over the peptide backbone, attempting to force a trajectory that would leave the side chain solvent exposed along the pathway. Unlike the under-peptide case, in these simulations the peptide would not move into a TCR-bound state even with spring

**Fig. 6 | The flip in the neoantigen occurs via an under-peptide motional pathway resembling a limbo dance. a** Energy diagram describing the peptide conformational change as a transition from a lower to higher energy flipped state, resulting in a high energy barrier in the forward direction and a low barrier in the reverse direction. **b** Under-peptide motional pathway illuminated by the reverse WEMD simulations beginning with the conformation in the ternary complex with TCR4, showing pTrp6 of the neoantigen moving underneath the peptide backbone. The three under-peptide conformations are extracted from three roughly equally spaced time points of a single successful transition. **c** Solvent accessible surface area of the pTrp6 side chain from each frame of the 109 successful reverse WEMD transitions. The solid cyan-to-yellow curve was generated from LOWESS smoothing of the data. The value expected from an exposed side chain if pTrp6 transitioned by moving over the top of the peptide (-120 Å$^2$) is indicated by the arrow. **d** RMSD of pTrp6 from the TCR4-bound conformation in SMD simulations of the peptide/HLA-A3 complexes forcing under- or over-peptide rotations. Data are shown for both the neoantigen and the WT peptide as a function of progressively larger spring constants. Only the neoantigen in an under-peptide rotation reaches the bound state, as indicated by the top row. **e** Number of unfavored ϕ/ψ torsion angles for all non-terminal amino acids of the neoantigen in the under- or over-peptide SMD simulations. Attempting to force an over-peptide rotation by increasing the spring constant results in greater torsional resistance. **f** Distance between the center of mass (COM) of the pTrp6 side chain and the position 2 side chain in the WT peptide (left) or neoantigen (right) under-peptide SMD simulations, colored by degree of van der Waals (vdW) overlap between the side chains. In the neoantigen simulation, the side chains remain distant, with little to no atomic overlap. In the WT peptide simulation, the side chains come in close proximity, with substantial overlap as the simulation progresses. Data are from the simulations with the 100 kJ/mol/mm$^2$ spring constant. Bracketed values give the average atomic overlap in Å.

constants as high as 1600 kJ/mol/nm$^2$ (Fig. 6d, neo over). The closest conformation reached still possessed an RMSD more than 6 Å from its conformation in the TCR4 ternary complex (Supplementary Fig. 8C). To address why there was no successful transition, we asked whether over-peptide rotation might be torsionally unfavored due to constraints on peptide ϕ/ψ bond angles. We computed ϕ/ψ torsion angles along the peptide backbone for both over- and under-peptide rotations and for all spring constants simulated, noting any instance of unfavored torsions (i.e., ϕ/ψ angles outside of the allowed and generously allowed regions of the Ramachandran plot for all non-terminal residues of the peptide). For the over-peptide simulations, we found that torsional strain increased dramatically with increasing values of the spring constant (Fig. 6e). This finding indicates that adding additional torque to try to force an over-peptide transition yields greater resistance from torsional constraints. In contrast, no such association between torque and unfavored ϕ/ψ torsions was noted for the under-peptide simulations. While fewer unfavored torsions were observed for the over-peptide simulations at low torques, this is explained by the simulation's resistance to adopting unfavored torsions in the absence of large forces. As torque increases to overcome this resistance, even more torsional strain is encountered relative to the under-peptide simulations, and the peptide still does not flip.

### The limbo dance flipping mechanism is sterically hindered in the WT peptide due to the motions in the WT peptide/HLA-A3 complex

We next used the same SMD procedure that illuminated the flip in the neoantigen to examine rotation of the WT peptide, beginning with the structure of the TCR-free peptide/HLA-A3 complex and applying force to try and rotate the pTrp6 side chain into the position seen when bound to TCR4. We could not force an under-peptide transition, even with spring constants as high as 1600 kJ/mol/nm$^2$ (Fig. 6d, WT under). Investigating why, we found that the higher mobility of the pHis2 side chain allowed it to move into the path of the rotation, sterically blocking the transition due to the formation of contacts and even van der Waals overlap between atoms of the pTrp6 and pHis2 side chains (Fig. 6f, left). This was not seen with the neoantigen, as the less mobile pLeu2 remained distant from the pTrp6 side chain (Fig. 6f, right). These results are consistent with the unbiased traditional MD simulations, which for the WT peptide but not the neoantigen showed contacts and the potential for steric interference as pTrp6 entered the base of the groove (Fig. 5f, g). As with the neoantigen, we also could not force the WT peptide to flip via an over-peptide rotation (Fig. 6d, WT over).

### Experimental confirmation of the allosteric influence of the position 2 anchor on the motions of pTrp6 in the HLA-A3 binding groove

The molecular dynamics simulations indicate that our inability to detect TCR recognition of the WT peptide/HLA-A3 complex emerges from how the position 2 amino acid influences the motions of the peptide in the HLA-A3 binding groove, with the WT sample possessing greater dynamic behavior but a hindered ability to transition underneath the peptide backbone to the TCR-bound flipped state. To confirm that the identity of the position 2 amino acid indeed alters peptide motions, we used $^{19}$F nuclear magnetic resonance (NMR) spectroscopy, replacing the pTrp6 in the neoantigen and WT peptide with 5-fluoro-tryptophan (5F-Trp), which in the static crystallographic structures of the neoantigen and WT peptide/HLA-A3 complexes would leave the fluorine atom solvent exposed (Supplementary Fig. 9A). The fluorine chemical shift is highly sensitive to the local environment due to the paramagnetic shielding from the electrons of the fluorine atom[41–43]. Although structural interpretations of protein fluorine spectra are notoriously difficult, comparative analyses can yield insight into conformational and dynamic differences[44,45].

Anticipating lengthy NMR data collection times, we first verified the stability of the peptide/HLA-A3 complexes. We previously used variants of the neoantigen and WT peptides fluorescently labeled at position 5 to monitor peptide dissociation rates via fluorescence anisotropy[15]. We repeated those experiments here but expanded the temperature range to include 25 °C and 4 °C in addition to our previously reported 37 °C data. Although the WT complex was less stable at 37 °C and 25 °C, stability was enhanced at 4 °C, with only a small amount of dissociation observed over an extended period (Supplementary Fig. 9B). This result gave us confidence to proceed with NMR experiments, but still cautioned us about the potential for sample degradation as described below. To verify that the presence of the fluorine atom does not fundamentally alter the behavior of the system, we investigated TCR binding to the 5F-Trp substituted neoantigen/HLA-A3 complex. Using SPR, we measured a $K_D$ for TCR3 that was very close to that measured with the unlabeled neoantigen complex, indicating that fluorine on pTrp6 does not alter the ability of the peptide to flip or significantly destabilize the TCR-bound state (Supplementary Fig. 9C).

We examined one-dimensional $^{19}$F spectra of freshly purified samples of HLA-A3 presenting either the neoantigen or wild-type peptide. All data were collected at 5 °C to promote sample stability. $^{19}$F NMR spectra of the free peptides under the same conditions indicated the chemical shifts of the fully hydrated fluorine in the free peptides, which were −125.05 ppm and −125.12 ppm for the neoantigen and WT peptide, respectively (Fig. 7a, e). The spectrum for the HLA-A3 complex with the neoantigen featured an upshifted major peak at −125.18 ppm (Fig. 7a). The fluorine signals from the complex were significantly broader than those of the free peptide, indicating that the $^{19}$F atom experienced slow rotational diffusion as expected for a 44 kDa peptide/HLA-A3 complex. Closer examination of the spectrum revealed two additional minor peaks at −123.09 ppm and −126.90 ppm with the total signal areas split among the three peaks as 15%, 77%, and 8% (Fig. 7b). We interpreted the major peak at −125.18 ppm as originating from a fluorine accessible to solvent yet tightly associated with

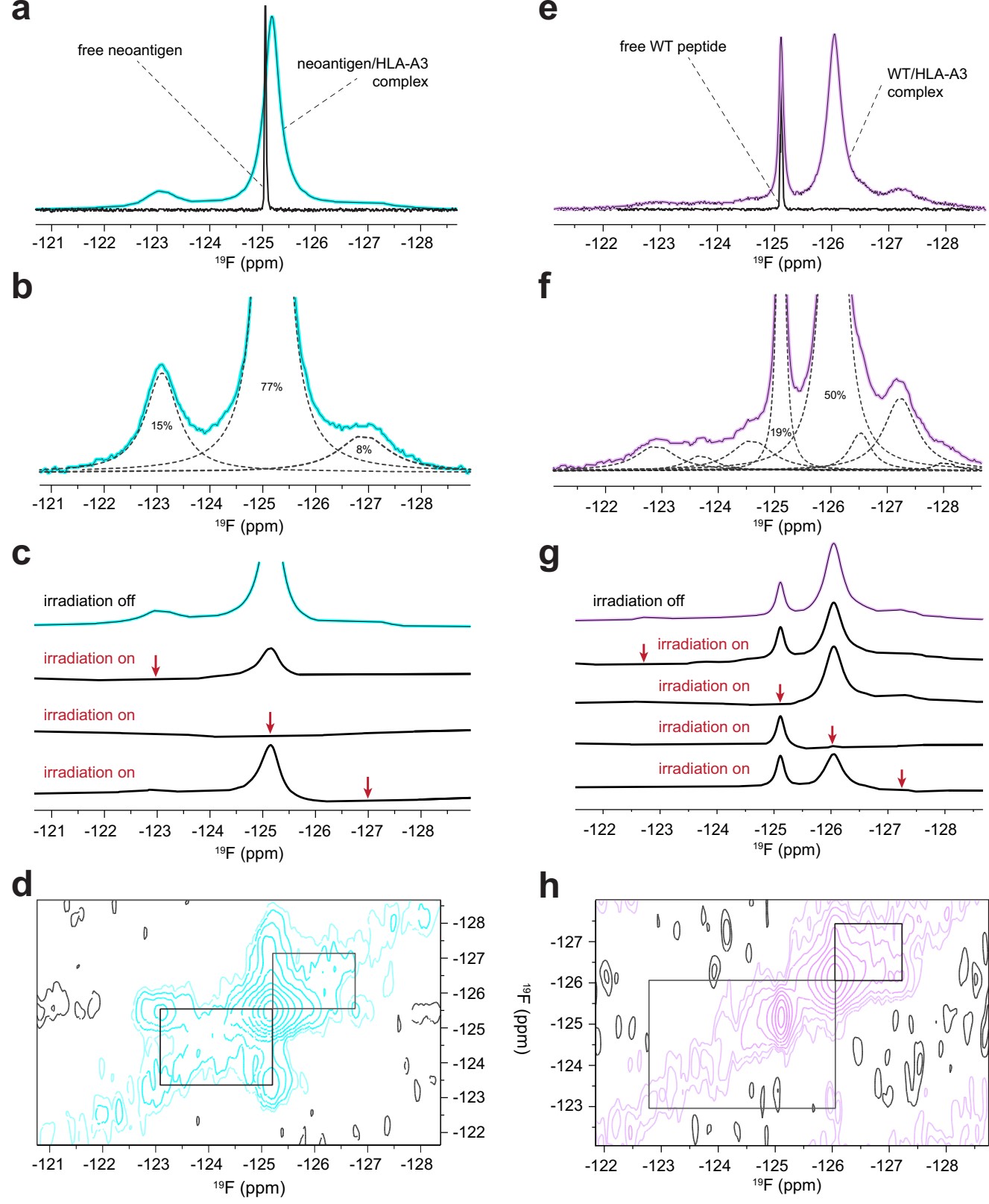

the protein. The two minor peaks at −123.09 ppm and −126.90 ppm were also broadened, indicating two additional states for the complex with distinct environments for the fluorine atom.

To establish that the three peaks in the neoantigen complex reflected interconverting conformations of the pTrp6 side chain, we performed chemical exchange saturation transfer (CEST) experiments.

These experiments confirmed that all three states in the neoantigen/HLA-A3 complex were in slow conformational exchange, as selective pre-saturation of any of these three peaks significantly reduced the intensity of the remaining two signals (Fig. 7c). Two-dimensional exchange spectroscopy (EXSY) experiments independently confirmed that all three peaks in the neoantigen complex originated from the

**Fig. 7 | Experimental confirmation of differential peptide dynamics in the HLA-A3 binding groove through ¹⁹F NMR. a** One-dimensional ¹⁹F NMR spectra of the free neoantigen and the neoantigen/HLA-A3 complex. Whereas the spectrum of the free peptide shows a single sharp peak, the spectrum of the complex shows the presence of multiple states, with broad linewidths as expected for the 44 kD complex. Note that the resonance frequency of the free peptide and the major form of the neoantigen/HLA-A3 complex do not coincide. **b** Zoomed in view of the neoantigen/HLA-A3 complex, with relative peak areas as determined by line shape fitting (dashed lines). **c** Conformational Exchange Saturation Transfer (CEST) experiments demonstrating that the multiple peaks in the neoantigen/HLA-A3 complex result from the ¹⁹F spin experiencing a slow dynamic exchange between at least three different conformations. All four traces were acquired with the same receiver gain, and positions of selective irradiation are shown by red arrows. Irradiation of each of the major resonances significantly reduced the intensity of the two other resonances indicating that they arise from the same ¹⁹F spin dynamically switching between distinct environments. **d** Two-dimensional Exchange Spectroscopy (EXSY) experiments independently confirm all three peaks of the neoantigen complex are in conformational exchange. Black rectangles indicate positions of cross-peaks expected if the diagonal peaks represented alternative environments of ¹⁹F in dynamic equilibrium in a slow exchange regime. The mixing time of the experiment was 50 ms. **e** As is in panel (**a**) but for the free WT peptide and the WT

peptide/HLA-A3 complex. Note the coinciding but different widths of the resonance of the free peptide and one of the two major peaks for the peptide/HLA-A3 complex, as well as the more complex pattern of additional peaks for the WT complex compared to the neoantigen complex shown in panel (**a**). **f** As in panel (**b**) but zoomed in for the WT peptide/HLA-A3 complex. Line shape fitting reveals at least eight peaks, compared to the three with the neoantigen complex in panel (**b**). Excluding the sharper peak at −125.12 ppm, the linewidths of these peaks are similar to those of the neoantigen complex and consistent with a 44 kD complex. **g** As in panel (**c**) but for the WT peptide/HLA-A3 complex. Selective irradiation at each resonance reduced the intensity of the others except for the resonance at −125.12 ppm. Likewise, selective irradiation at −125.12 ppm saturated this resonance but did not alter the intensity of the other peaks. The multiple broad peaks thus represent states in slow conformational exchange on the NMR time scale with exception of the −125.12 ppm resonance, which corresponds to a distinct, non-exchanging but still protein-bound population of the peptide. **h** As in panel (**d**), but for the WT peptide/HLA-A3 complex, showing cross-peaks for major resonances except for that at −125.12 ppm. The lower signal-to-noise ratio in the WT peptide/HLA-A3 sample led to poorer detection of some cross-peaks, yet still confirms conformational exchange between at least three conformational states as well as the non-exchanging character of the −125.12 ppm resonance.

same molecule transitioning between three conformational states, as indicated by cross-peaks connecting the resonance frequencies of the exchanging conformations (Fig. 7d).

By contrast, the spectrum for the WT peptide/HLA-A3 complex was more complex than that of the neoantigen. There were two major peaks, the sharper of which at −125.12 ppm aligned with the position of free peptide and represented about 19% of the total signal (Fig. 7e). A second broader peak was observed upfield at −126.04 ppm and corresponded to roughly 50% of the remaining population. Line shape fitting revealed that the spectrum for the WT peptide/HLA-A3 complex contained several additional peaks from −122.90 ppm to −128.03 ppm (Fig. 7f). The upshifted major peak is consistent with reduced exposure of the ¹⁹F nucleus to the base of the binding groove[46], as suggested by our MD simulations, while the greater number of peaks indicated a larger conformational ensemble than seen with the neoantigen. CEST experiments showed that all the peaks were in exchange except for the peak at −125.12 ppm, as selective pre-saturation influenced all but the −125.12 ppm signal, and irradiation at −125.12 ppm did not influence the others (Fig. 7g). This result was confirmed by two-dimensional EXSY, which revealed a lack of detectable cross-peaks with the −125.12 ppm signal (Fig. 7h). As it was sharper and not in exchange, we interpret the peak at −125.12 ppm as resulting from sample degradation, owing to the lower stability of the WT peptide/HLA-A3 complex. We confirmed this by re-collecting data on the same sample after ~11 months of storage at 4 °C. The spectrum of this aged complex showed a significant increase in the resonance at −125.12 ppm and a corresponding decrease in the others, along with the emergence of a new resonance at −124.92 ppm (Supplementary Fig. 9D). The fact that the major degradation peak at −125.12 ppm overlapped with but was broader than the signal of the free peptide indicates that in the degraded state, the fluorine atom experiences a local environment similar to that in the free peptide, yet the peptide is tumbling slowly and likely still associated with HLA-A3 heavy chain, potentially as part of a soluble misfolded HLA-A3 heavy chain[47].

The spectrum of a similarly aged neoantigen/HLA-A3 complex was less distorted but did show the emergence of peaks at −124.92 ppm and −125.12 ppm, indicating a similar but less populated degradation pathway, as would be expected given the higher stability of the neoantigen/HLA-A3 complex (Supplementary Fig. 9E). Overall, its lower stability notwithstanding, the WT peptide/HLA-A3 complex shows more dynamic complexity than that of the neoantigen, confirming that the identity of the position 2 amino acid influences motional properties in the center of the peptide.

## The pTrp6 side chain is rigidified in the TCR-bound ternary complex

Although the experiments with the Bta-modified peptide confirm the importance of the flip in pTrp6 for TCR binding revealed by the static structures, ¹⁹F NMR provides the opportunity to experimentally examine how the tryptophan dynamics change upon TCR binding. We thus generated a sample of the ternary complex of TCR3 bound to the neoantigen/HLA-A3 complex with 5F-Trp at position 6, with the TCR at high enough concentration to yield a sample with ~90% of the neoantigen/HLA-A3 complex bound to the TCR (2.25 mM TCR and 0.5 mM neoantigen/HLA-A3; TCR3 was selected for this experiment as it was easier to produce in the quantity required). A one-dimensional ¹⁹F spectrum collected at 5 °C for this sample revealed a spectrum with only two quantifiable peaks: a major, broad peak at −127.42 ppm with a sharper shoulder near −125 ppm (Fig. 8a). The shoulder overlaps with the main peak seen in the spectra of the free neoantigen/HLA-A3 complex and likely reflects the fraction of unbound peptide/MHC in the sample (our inability to observe the other peaks seen in the free neoantigen/HLA-A3 complex can be attributed to the small percentage of unbound protein). The major peak for the ternary complex is at a chemical shift not seen in the spectrum of the free peptide/MHC complex. The broad line width of this peak is consistent with the formation of a slowly tumbling, cylindrically shaped TCR3-neoantigen/HLA-A3 complex. That only a single symmetrical peak was observed for the complex is consistent with a restrained side chain adopting one major conformation, and its position in the spectrum indicates a chemical environment for the ¹⁹F nucleus distinct from that in the free neoantigen/HLA-A3 complex, as anticipated from the crystallographic and simulation data.

To further interpret these results, we performed unrestrained atomistic MD simulations of the neoantigen ternary complexes with TCR3 and TCR4 and examined the motions of pTrp6, simulating each complex for 1 μs in explicit solvent. From these simulations, we examined the volume sampled by pTrp6. Consistent with the NMR data, the side chain was rigid, with density visible for the atoms of the tryptophan side chain (Fig. 8b; compared with Fig. 4d, e). We repeated these simulations two additional times, yielding average RMSFs for pTrp6 in the ternary complex that were ~60% of those in the free neoantigen/HLA-A3 complex (Supplementary Fig. 10A). This same analysis showed that pLeu2 remained similarly rigid upon TCR binding (Supplementary Fig. 10B). TCR binding to the neoantigen/HLA-A3 complex thus rigidifies the flipped pTrp6 side chain, without altering the motions of pLeu2.

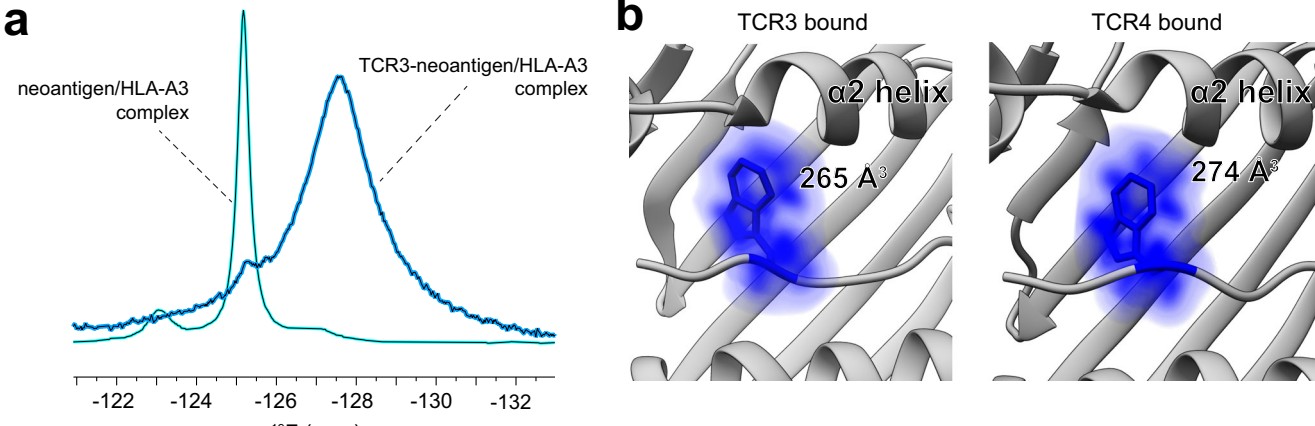

**Fig. 8 | TCR binding rigidifies the flipped pTrp6 of the neoantigen. a** One dimensional [19]F NMR spectra of the neoantigen in the TCR3-neoantigen/HLA-A3 ternary complex, along with the data from the unbound neoantigen/HLA-A3 complex for comparison (see Fig. 7a). The peak for the ternary complex is extensively broadened, as expected from its slower tumbling rate. A single shoulder aligns with the major peak of the unbound complex, reflecting the small percentage of neoantigen/HLA-A3 that is unbound in the sample. The presence of a single resonance at a different position for the TCR-bound state indicates binding-associated rigidification of the flipped pTrp6 side chain. **b** Molecular dynamics simulations of the TCR-neoantigen/HLA-A3 ternary complexes illustrate the rigidification of the flipped pTrp6 upon TCR binding. Results show conformational space occupied by pTrp6 during 1 μs simulations of the ternary complexes with TCR3 and TCR4. Color density reflects degree of sampling (voxels sampled <10% of the time are excluded), values give volumes of sampled space.

## Discussion

A defining feature of TCRs is their inherent cross-reactivity, a biological necessity due to the relatively T cell repertoire relative to the large universe of potential epitopes. Cross-reactivity in turn emerges from the structural features of both the TCR and its peptide/MHC ligand, as well as the weak-to-moderate binding affinities that characterize TCRs post-thymic selection. Long-held estimates place the average number of peptide/MHC ligands that are compatible with a given TCR as high as a million[1,2]. Remarkably though, TCRs can also show very high specificity, detecting even subtle changes in the composition of peptides. While this can often be explained using well known structure/activity relationships (for example, changes to peptides that alter hotspots in the interface or significantly change peptide conformation)[3], TCRs often show surprising and unpredictable sensitivity to peptide modifications that cannot be easily rationalized, even when high-resolution structural information is available. This sensitivity complicates numerous efforts in immunology that involve TCR specificity, including the selection and engineering of vaccine candidates, predicting immunogenic epitopes, understanding mechanisms of escape mutations, and selecting and optimizing TCRs for potential immunotherapies.

Here we studied the remarkable capacity for TCRs to distinguish between optimal and suboptimal primary anchors in a public neoantigen resulting from a recurrent hotspot mutation in the driver oncogene *PIK3CA*. TCR sensitivity to anchor modification, separate from the impacts on peptide binding to MHC, is well-recognized[6,8–13]. In prior work, we suggested that differences in peptide anchor residues could modulate the motional properties of the peptide and MHC binding groove, impacting TCR recognition via dynamic allostery, in which protein motions propagating from changes at one site influence the behavior at another site[48,49]. Although difficult to demonstrate as it often manifests without obvious changes in static structures, dynamic allostery has emerged as a fundamental component of protein behavior, influencing molecular recognition and signaling throughout biology[50]. While the sensitivity of the motions of peptide/MHC complexes to peptide modifications is well established[10,51–54], clear demonstration of peptide/MHC dynamic allostery modulating TCR specificity has been lacking.

Our results show unambiguously that dynamic allostery controls the specificity of TCRs to a mutation-induced anchor modification in the PI3Kα neoantigen presented by HLA-A3. The mechanism relies on motions intrinsic to the peptide/HLA-A3 complexes, with rapid movements of the histidine at position 2 of the WT peptide hindering a large and much slower conformational flip in the tryptophan at position 6 that is required for TCR binding. The hindrance from the WT peptide evokes a dynamic gate and only emerges because the path the tryptophan 6 must follow as it transitions across the HLA-A3 peptide binding groove takes it underneath the peptide backbone. In the WT peptide, high-frequency motions resulting from the suboptimal position 2 anchor interfere with this process. The path the tryptophan takes is in turn facilitated by the presence of a large cavity underneath the peptide backbone, which the neoantigen but not the WT peptide is able to sample.

Although our structural and NMR experiments, along with the experiments with the Bta-modified neoantigen demonstrate the importance of the flip in the tryptophan, an unresolved mechanistic question is why the large conformational change is required for TCR binding in the first place. One clue is found in the conformations sampled: while the tryptophan in the unbound neoantigen/HLA-A3 complex mostly samples space on the side and in the base of the groove, it has a propensity to reach up and out, where it would be incompatible with an incoming TCR. As the TCR closes in, attracted by complementary features distributed across the surface of the peptide/HLA-A3 complex[55–57], available space above the peptide groove will be reduced and water expelled. We hypothesize that this reduction and expulsion of solvent destabilizes the relatively polar tryptophan as it rests alongside the α1 helix, and it thus takes advantage of the space under the peptide backbone and rotates underneath and around to the α2 helix side of the groove, where it can form new stabilizing electrostatic interactions that help offset the energetic cost of the flip. A partial flip, with the tryptophan remaining in the base of the groove, is seemingly unable to be stabilized. Related to this, a hypothetical over-peptide flip, beyond being incompatible with peptide torsions as we show, would be incompatible with the initial and transitory TCR-peptide/HLA-A3 contacts formed during the TCR binding process.

This proposed mechanism for the conformational flip in the neoantigen is relevant to the panel of TCRs we have studied; the extent to which it might apply to other receptors specific for the PI3Kα neoantigen remains unknown. In many cases, different TCRs specific for the same target bind similarly[15,58–61]. However, in other cases,

different TCRs bind differently to the same target[62–67]. A hypothetical PI3Kα neoantigen-specific TCR that binds in a manner that avoids the conformational sampling of pTrp6 of the peptide may not necessitate such a conformational change. Although we believe this would be unlikely given the position of pTrp6 in the center of the peptide, in that hypothetical case, any selectivity between the neoantigen and WT peptide could emerge from more subtle structural or dynamic differences between the peptide/MHC complexes, potentially extending into force-induced changes in protein dynamics that are becoming recognized as additional contributors to TCR specificity as discussed below.

Our results do not directly address the difference in the in vivo immunogenicity of the PI3Kα neoantigen compared to its corresponding WT peptide, although they provide important clues. As noted above, TCRs that are not susceptible to the differential dynamics between the neoantigen and its WT counterpart could exist. However, the fact that the two peptides are antigenically distinct (at least with the receptors studied) may enable tolerance mechanisms to discern them, resulting in T cell repertoires that are more responsive to the neoantigen than to the WT peptide[68]. The weaker binding of the WT peptide to HLA-A3 likely also contributes to its apparent lack of in vivo immunogenicity, although we note that in multiple instances, neoantigens that bind very weakly to class I MHC proteins have been shown to be potent drivers of tumor immunity provided they are antigenically distinct from their WT counterparts[69–71]. With growing efforts to identify immunogenic cancer neoantigens, understanding the interplay between how mutations in peptides alter TCR recognition *versus* how they alter peptide-MHC binding is becoming more important.

While the mechanistic details uncovered here are specific to TCR discrimination between the PI3Kα neoantigen and its WT counterpart with select TCRs, multiple aspects of our findings are applicable to TCR specificity in general. As noted above, TCR sensitivity to subtle peptide modifications in the absence of structural changes is well documented, occurring in viral, self, and tumor antigens[6,9–13]. The varied motions of peptides in MHC binding grooves have been shown by multiple experimental techniques[72,73], as has the ability of different peptides to alter the dynamics of MHC proteins[10,51–54]. As we quantify here, conformational changes in peptides upon TCR binding are commonly seen, as are open volumes within the peptide-binding groove. We thus suggest that dynamic allostery is common in TCR recognition of peptide/MHC, emerging from the architecture of peptide/MHC complexes (particularly for class I proteins) and the dynamic nature of proteins in general. We suggest that this process, which notably occurs within the ligand of the TCR, is a key component of specificity in T cell recognition, reflecting the cooperative evolution of a receptor-ligand sensing system that must simultaneously possess cross-reactivity and specificity.

Similarly, there is a growing appreciation of mechanosensing in T cell biology and a role for applied forces in influencing TCR specificity, the latter via force-induced changes in the lifetimes and structural properties of membrane-bound TCR-peptide/MHC complexes[74–77]. These manifest through changes in the energetic and thus dynamic landscapes of protein complexes under applied force[74,78,79]. Although force-dependent T cell responses are not needed to explain specificity in this case, the ability of peptide modifications to allosterically modulate peptide/MHC dynamics could in turn alter force-induced changes in protein motions, thus influence lifetimes and signaling. This possibility could potentially contribute to the accumulating number of instances where T cell specificity does not correlate with TCR binding properties measured in solution.

An aspect that remains unclear is the extent to which evolution has tuned the capacity for dynamic allostery in peptide/MHC complexes. Although the many thousands of class I MHC variants are structurally homologous, the roles of the underlying MHC polymorphisms distributed throughout the peptide-binding groove have

not all been clarified[80]. While their impact on peptide selection and other features such as protein stability is well appreciated[81], in other cases, they have also been shown to directly or indirectly impact TCR recognition, not always in structurally visible ways[82–88]. It is thus possible that evolution has selected for polymorphisms that tune MHC energy landscapes, allowing for differential dynamic responses to peptides and further enhancing the diversity and complexity of antigen presentation and recognition in cellular immunity.

## Methods

### Recombinant protein preparation

The PI3Kα neoantigen, wild-type peptide, Bta, 5F-Trp, and 5-carboxyfluorescein-modified lysine peptide variants were purchased from GenScript at >80% purity and dissolved in DMSO prior to refolding. The Bta and 5F-Trp amino acids are indexed at PubChem (ID numbers: Bta 150953; 5F-Trp 24721259). Proteins, including peptide/MHC complexes, TCRs, and the s3-4 scTv were purified from bacterially expressed inclusion bodies[15]. Briefly, the extracellular domain of the MHC heavy chain, β2-microglobulin, TCR α and β chains, and the s3-4 scTv were overexpressed in *Escherichia coli* and the resulting inclusion bodies solubilized in 8 M urea and 6 M guanidinium-HCl. Denatured proteins were refolded in either peptide/MHC refolding buffer (400 mM L-arginine, 100 mM Tris-HCl, 2 mM $Na_2EDTA$, 6.3 mM cysteamine, 3.7 mM cystamine and 0.2 mM PMSF, pH 8.3) or TCR refolding buffer (50 mM Tris-HCl, 2.5 M urea, 2 mM $Na_2EDTA$, 6.5 mM cysteamine, 3.7 mM cystamine and 0.2 mM PMSF, pH 8.15) at 4 °C and incubated overnight. The refolding buffer was then dialyzed against ultrapure $H_2O$ followed by 10 mM Tris-HCl (pH 8.3) at room temperature (for peptide/MHC) or 4 °C (for TCR and scTv) for 48 h. The refolded proteins were subsequently purified by anion exchange followed by size exclusion chromatography. Protein concentrations were determined by UV absorbance using sequence-determined extinction coefficients.

### X-ray crystallography

The neoantigen/HLA-A3 and Bta-substituted neoantigen/HLA-A3 complexes were solubilized in 10 mM HEPES, 20 mM NaCl, pH 7.4 prior to crystallization. Crystals were obtained by hanging drop vapor diffusion at 4 °C. Crystals of the replicate neoantigen/HLA-A3 complex grew from 12% w/v polyethylene glycol 3350 and 4% v/v Tacsimate (Hampton Research HR2-827) at pH 6.0. Crystals of the Bta-substituted complex grew from 15% w/v polyethylene glycol 3350 and 150 mM CsCl. Crystals were cryoprotected with 15–25% glycerol prior to flash-freezing in liquid nitrogen. X-ray diffraction data were collected at the 23-ID-D (neoantigen complex) and 24-ID-C (Bta-substituted neoantigen complex) beamlines of the Advanced Photon Source at Argonne National Laboratory. Diffraction data were processed through HKL2000[89] and initially phased by molecular replacement using Phaser in Phenix[90]. Space group determinations were confirmed by processing the data through DIALS followed by merging and scaling with Aimless in CCP4[91]. Search models for both complexes were PDB 7L1C with the peptide removed. Peptides were then manually rebuilt in Coot after initial models were obtained from Phenix AutoBuild. Models were further refined automatically in Phenix and manually in Coot[92]. Composite/iterative build OMIT maps were calculated with simulated annealing using CNS as implemented in Discovery Studio 2023. Structures were visualized and analyzed in PyMOL and Discovery Studio.

### Structural analyses and comparisons

Solvent accessible surfaces and surface areas were calculated in either Discovery Studio or VMD[93] with a 1.4 Å radius probe. $pK_a$ calculations using continuum electrostatics were performed with H++ using default options[94]. The $pK_a$ of pHis2 in the WT peptide bound to HLA-A3 was calculated using the WT peptide/HLA-A3 structure (PDB 7L1B). The $pK_a$

of pHis2 in the free WT peptide was calculated using the conformation of the peptide from PDB 7L1B with the atoms of the heavy chain and $\beta_2$-microglobulin removed. The $pK_a$ of pHis3 in the WT peptide and neoantigen bound to HLA-A3 were calculated using the previously published structures (PDB 7L1B and PDB 7L1C) and the second neoantigen/HLA-A3 structure reported here (PDB 8VCL); for pHis3 in the neoantigen, the values from the two structures were averaged. Electrostatic destabilization was calculated from $pK_a$ values using $\Delta\Delta G° = -2.303RT(pK_{a,bound} - pK_{a,free})$, where $R = 1.987$ cal/K/mol and $T = 298.15$ K. Cavities in peptide/MHC complexes were quantified with CAVER Analyst 2.0 with default options except pockets were excluded, recording the summed volumes of the largest contiguous pockets between the peptide and the HLA-A binding groove[95]. Calculations were performed on peptide/MHC complexes in the A3 superfamily in the PDB as of October 15, 2024. Complexes with cleaved, covalently bonded, or non-peptidic ligands; with TCRs or other proteins bound; or with peptides with non-natural amino acids were excluded. MoloVol 1.1 was used for visualization of the cavities in Supplementary Fig. 7 [96]. TCR-free/TCR-bound RMSD calculations for nonameric peptides presented by class I MHC proteins were calculated via UCSF Chimera[97], identifying all pairs of identical nonamers in the PDB as of March 24, 2024. For instances in which peptides were represented by multiple, replicate structures, we only included the comparison that yielded the largest Cα or all atom RMSD. In cases where there was a numerical mismatch of atoms between pairs of peptides due to unresolved atoms, atoms were stripped from the fully resolved peptide to match the peptide with missing atoms. As structures exist for some peptides in complex with multiple TCRs, the analysis yielded 67 comparisons for 39 nonameric peptides. For both RMSD and cavity calculations, when multiple molecules were present in crystallographic asymmetric units, only the first molecules were used. Structural comparisons involving the TCR-free neoantigen/HLA-A3 structure used the higher resolution structure (PDB 7L1C) except where otherwise indicated.

## Binding measurements
Affinities and binding kinetics were measured via SPR using a Biacore T200 instrument. Proteins were buffer exchanged into HBS-EP buffer (10 mM HEPES, 150 mM NaCl, 3 mM EDTA, 0.005% surfactant P20, pH 7.4; Cytiva BR100669) prior to experiments. The s3-4 scTv was employed as a positive control and the irrelevant Tax$_{11-19}$/HLA-A2 or gp100$_{209}$/HLA-A2 complexes as negative controls. For affinity measurements, TCRs and s3-4 were immobilized on a CM5 Series S sensor chip to 400-6000 RU via amine coupling and peptide/MHC complexes were injected at a flow rate of 5 μL/min. Experiments were performed at either 4 °C or 25 °C as indicated with a blank activated and deactivated flow cell as reference. Binding affinities were determined by fitting the curves of the reference-subtracted steady-state responses against the injected protein concentrations to a 1:1 binding model in OriginPro 2024. In most cases injections at each concentration were repeated twice and both values fit simultaneously for a single measurement. For measurements of TCR dissociation rates ($k_{off}$ values), experiments were performed at 4 °C. TCRs or the s3-4 scTV were immobilized to ~300 RU on a CM5 Series S sensor chip with peptide/MHC complexes injected at a flow rate of 100 μL/min. Dissociation rates were determined by fitting the dissociation phase of a sensorgram to a single exponential decay function in OriginPro 2024. Association rates ($k_{on}$ values) were calculated from the ratio of the measured $k_{off}$ and the separately measured $K_D$ at 4 °C using $k_{on} = k_{off}/K_D$.

## T cell transduction and functional analysis
For TCR-T generation, the 293GP cell packaging line (Takara Bio 631530) was plated overnight onto poly-D-lysine coated 60 mm² plates at $1.6 × 10^6$ cells/ plate in complete DMEM (Corning 10013CV). 293GP cells were transfected with 6 μg of TCR3 or TCR4 pMSGV1-plasmid along with 3 μg of the RD114 envelope using the Lipofectamine 3000 reagent set (Invitrogen L3000001). The pMSGV1 retroviral plasmid and RD114 plasmid were obtained through an MTA from S. A. Rosenberg (NCI, Bethesda, USA). Viral supernatant was collected after 48 h and loaded onto retronectin-coated (10 μg mL$^{-1}$; Takara Bio T110A) non-TC 24-well plates and centrifuged at $2000 × g$ for 2 h at 32 °C. T cells enriched from PBMCs from healthy donors (STEMCELL Technologies 200-0092) were stimulated with TransAct (Miltenyi Biotech 130-111-160) and rhIL-2 (Biotechne BT-002-AFL-050; 30 IU/mL) for 48 h prior to transduction. Stimulated cells were plated at $(1–5) × 10^5$ cells/ well and spinoculated at 1500 rpm for 15 min and placed at 37 °C until ready to use. TCR-T cells were utilized for downstream functional analysis at days 5–8 post transduction. Use of human PBMCs was reviewed and approved by the Memorial Sloan Kettering Cancer Center Institutional Review Board (IRB; protocol no. 17-250). PBMCs from STEMCELL Technologies were obtained under informed consent from volunteers, approved by an IRB from WCG Clinical Services.

The peptide specificity of TCR-transduced T cells was assessed via intracellular cytokine staining using the BD Cytofix/CytoPerm Plus Kit (BD 554722), following the manufacturer's instructions. Cos-7 cells were electroporated with 100 μg/mL of *HLA-A*03:01* mRNA and plated into 96-well round-bottom plates overnight. Cells were pulsed with titrating amounts of indicated peptides for 30 min at 37 °C. Cells were washed with 1× PBS to remove any unbound peptide. TCR-expressing T cells were co-cultured at an E:T ratio of 1:1 for 6 h in the presence of anti-CD107A-BV650 (Clone H4A3, BioLegend) and Golgi block (BD 554724). Cells were washed in 1× PBS and surface labeled with Live/Dead fixable dye (Invitrogen), anti-CD3-APC-H7 (Clone SK7, Invitrogen), anti-CD8-eFluor450 (Clone SK1, Invitrogen) and anti-mouse TCR-PerCpCy5.5 (Clone H57-597, Invitrogen) for 30 min at 4 °C. Cells were washed again with PBS and then fixed and permeabilized for 15 min at 4 °C. Surface-labeled cells were then washed with perm-wash buffer and labeled with anti-TNFα-PE (Clone Mab11, Invitrogen) for 30 min at 4 °C in perm-wash buffer. All antibodies were used at a final concentration of 5 μg/mL. Finally, cells were washed with perm-wash buffer, suspended in 2% FBS in PBS, and data acquired on an X20 LSR Fortessa flow cytometer with the BD FACSDiva software. Data were analyzed using FlowJo software version 10.6.2. Representative gating strategy is shown in Supplementary Fig. 11. A table of all antibodies used and their dilutions is shown as Supplementary Table 3.

## Differential scanning fluorimetry
Thermal denaturation of peptide/HLA-A3 complexes was performed using differential scanning fluorimetry using a Prometheus NT.48 instrument (NanoTemper) monitoring intrinsic tryptophan fluorescence[17]. Briefly, 10 μL of neoantigen/HLA-A3 and Bta-substituted neoantigen/HLA-A3 at concentrations of 15 μM in HBS-EP buffer (10 mM HEPES, 150 mM NaCl, 3 mM EDTA, 0.005% surfactant P20, pH 7.4; Cytiva BR100669) were loaded into instrument capillaries. The temperature was scanned from 20 to 95 °C at a constant rate of 1 °C/min. Fluorescence at emission wavelengths of 330 nm and 350 nm was recorded, and the first derivative of the ratio of the fluorescence intensities was plotted vs. the temperature to generate the melting curve. Data were fit to bi-Gaussian functions using OriginPro 2024.

## Traditional molecular dynamics simulations
Fully atomistic, unrestrained molecular dynamics simulations of the peptide/HLA-A3 complexes (referred to as traditional MD) were performed on GPU hardware with Amber18 using the ff14SB force field and an SPC/E water model[98]. Structures of the PI3Kα WT peptide/HLA-A3 complex (PDB 7L1B) and neoantigen/HLA-A3 complex (PDB 7L1C, using these coordinates as they were from the highest resolution structure) were utilized as the starting coordinates for each simulation. System setup is summarized in Supplementary Table 4. Peptides were modeled with charged terminal residues. As indicated by its calculated

$pK_a$ of 4.9, the pHis2 of the WT peptide was simulated in its uncharged state. pHis3 of the peptides was also left uncharged as discussed below. Sodium ions were added as counterions to neutralize the system. After initial energy minimization, systems were heated to 300 K with Langevin dynamics. Solute restraints were gradually relaxed under constant pressure from 25 to 0 kcal/mol/Å. 50 ps of NVT simulation was then performed, followed by production simulations. Production trajectories were calculated under constant volume with a 2 fs time step for a total time of 2 µs. Trajectories were analyzed with CPPTRAJ 18[99]. To confirm that the simulations had reached stable states prior to production, a time course analysis was performed in which various segments at the starts of the production runs were removed. The differences within the neoantigen and WT peptide/HLA-A3 complexes were maintained, indicating stable systems (Supplementary Fig. 12A).

Mass weighted amino acid RMSF values were calculated via the CPPTRAJ 'atomicfluct' command. 1D RMSD values were calculated via the CPPTRAJ 'rms' command after superimposition of the Cα atoms of the HLA-A3 binding groove (residues 1-180). D-score values were calculated using the differential φ/ψ angles of the simulation average peptide coordinates obtained via combinatorial usage of the CPPTRAJ 'average' and 'dihedral' commands[24,25]. Grid space occupancies of peptide residues were calculated via combinatorial usage of the CPPTRAJ 'bounds' and 'grid' commands using a grid spacing of 0.1 Å. Occupied grid space was visualized through the volume viewer in UCSF Chimera. The grid space volume was first smoothed via a Gaussian filter and contoured to encompass grid space occupied for at least 10% of simulation time. Side chain torsion angles and hydrogen bonds were calculated in CCPTRAJ using the 'dihedral' and 'hbond' commands, respectively. Conformational clustering of peptide residues was carried out in MATLAB R2022a and was performed on 2D-RMSD matrices calculated via the CPPTRAJ 'rms2d' command following Cα superimpositions of the HLA-A3 binding grooves (residues 1-180). The matrices were calculated in two different formats depending on the residue being investigated. For the position 2 anchor residue, because the side chain differs between the two peptides, independent matrices were generated for the neoantigen and WT simulations, each comprised of the simulation data and the respective initial crystallographic coordinates (PDB IDs 7L1B and 7L1C). For pTrp6, a single matrix was generated which was comprised of the simulation data of both the neoantigen and WT simulations, both sets of TCR-free crystallographic coordinates (PDB IDs 7L1B and 7L1C), and the crystallographic coordinates of the ternary complex with TCR4 (PDB ID 7L1D). The optimal number of clusters for each system of interest was calculated with the MATLAB 'evalclusters' command utilizing agglomerative clustering and the Calinski-Harabasz Index[100]. Dendrograms of the clustered 2D-RMSD data were generated via the MATLAB 'clustergram' command. Average structures of each cluster were generated via the CCPTRAJ 'average' command. Clusters were visualized by averaging the coordinates in the simulation frames representing each cluster. RMSD data in Fig. 4b and side chain contact counts in Supplementary Fig. 5C were smoothed using locally weighted scatterplot smoothing (LOWESS) as implemented in OriginPro 2024 using default options (unsmoothed data in Supplementary Fig. 13). Interatomic contacts were defined as interatomic distances <4 Å. Four independent replicate MD simulations were performed for the free peptide/HLA-A3 systems. As the replicate simulations all exhibited similar behavior (Supplementary Fig. 2A–C), grid space calculations and conformational clustering for the peptide/HLA-A3 complexes were performed using only the first sets of MD simulation data.

The calculated $pK_a$ of pHis3 in the peptide/HLA-A3 complexes was 5.8 for the WT peptide and 6.6 for the neoantigen (average value from both neoantigen structures). At physiological pH, this yields 2% of charged pHis3 for the WT peptide and 14% for the neoantigen. To investigate the potential impact of a minority population of the neoantigen complex with pHis3 protonated and charged, we ran

triplicate 2 µs simulations of the neoantigen/HLA-A3 complexes with pHis3 charged. The trend in flexibility across the peptide was unchanged (Supplementary Fig. 12B), and the analyses and subsequent simulations thus used neutral pHis3.

Simulations of the neoantigen/HLA-A3 complex bound to TCR3 and TCR4 were performed using the same preparation and production steps (using PDB IDs 7RRG and 7L1D as starting coordinates), except that the simulation time was 1 µs. Amino acids missing in the TCR3 structure were modeled using the neoantigen/HLA-A3 and ternary TCR structures as a template (PDB IDs 7L1C and 7L1D, respectively). Three independent replicate MD simulations were performed for each TCR complex simulated, all exhibiting similar behavior (Supplementary Fig. 10). Analysis of TCR complex simulations was also performed as described above.

### Weighted ensemble molecular dynamics simulations

WEMD of the neoantigen/HLA-A3 complex were performed via the Weighted Ensemble Simulation Toolkit with Parallelization and Analysis (WESTPA) package[37,38]. For forward simulations, starting coordinates were the neoantigen/HLA-A3 complex (PDB 7L1C), and the target was the position of pTrp6 in the TCR4-bound ternary complex (PDB 7L1D). This was reversed for the reverse simulation, with the coordinates of the TCR removed for the starting coordinates. The simulations were initiated from statistically independent configurations separated by 200 ns from traditional MD production simulations initiated and performed as described above. Full-atom RMSDs between pTrp6 in the neoantigen in the simulation and its position in the corresponding target coordinates were utilized as progress coordinates. The starting value of the progress coordinate was 6.4 Å for the forward simulation and 6.5 Å for the reverse simulation. Progress coordinates in both cases were divided into 72 bins, with steps designed to minimize bias in the transition directions recorded by WESTPA. Bin values were 0.0–2.0 Å (7 bins with an even step of 0.3 Å), 2.0–6.0 Å (40 bins with an even step of 0.1 Å), 6.0–11.0 (24 bins with an even step of 0.2 Å) and ≥11.0 Å (1 bin). Iteration time was set at 10 ps and the maximum number of trajectories in each bin was 8. All the simulation settings were the same as those for production simulations in traditional MD, except that distance restraints in the form of flat-welled parabolic potentials between opposing residues in the HLA-A3 α1 and α2 helices were applied to ensure that high energy states captured during WESTPA had proper binding groove geometry. Residues and the corresponding lower and upper distance bounds were Gly79-Ile142 (13.0–19.1 Å), Thr80-Ile142 (9.0–17.4 Å), Tyr59-Arg170 (10.0–18.3 Å), Tyr59-Asn174 (10.0–19.8 Å). Solvent accessible surface area data in Fig. 6c were smoothed using LOWESS as implemented in OriginPro 2024 using default options to generate the indicated curve. A hypothetical over-peptide conformation for computing the surface area in Fig. 6c was generated by manually adjusting the χ1 torsion of pTrp6 in PDB 7L1C to 27°, resulting in the side chain pointing directly out of the groove. For generating Supplementary Movie 1, the Cα atoms of the HLA-A3 peptide binding groove (residues 1-180) for all 5410 frames of a successful transition were aligned using CCPTRAJ, and the movie generated in Chimera with a step size of 20 frames.

### Steered molecular dynamics simulations

SMD was performed using enforced rotation[40]. Trajectories were generated using GROMACS 2022 with the CHARMM36-jul2021 force field[101,102]. Coordinates of the peptide/HLA-A3 complexes (PDB IDs 7L1B and 7L1C) were solvated with TIP3P water in a dodecahedral unit cell. System setup is summarized in Supplementary Table 4. All systems were charge neutralized by the addition of Na+ and Cl− ions to a final concentration of 150 mM and energy minimized via steepest descent. Short 500 ps of NVT ensemble dynamics were carried out in the presence of heavy atom restraints to stabilize the temperature to 298.15 K with the v-rescale thermostat. Systems were subsequently

equilibrated by 1 ns of NPT ensemble dynamics via c-rescale pressure coupling at 1 bar. Restraints were removed and production simulations initiated under the NPT ensemble utilizing Nose-Hoover and c-rescale temperature and pressure coupling, respectively. Rotation groups were defined as all atoms of the pTrp6 amino acid and rotation vectors selected along the peptide backbone. Enforced rotation was carried out in two directions (under and over) for a set of force constants using the flex2-t potential and rotation rate of 0.1°/ps for 500 ps. Force constants used were 100, 200, 400, 800, and 1600 kJ/mol/nm$^2$. For the neoantigen rotated under, the subsequent unbiased simulation adhered to the same simulation conditions except without enforced rotation parameters. RMSD measurements of rotated pTrp6 conformations relative to the TCR4-bound conformation (from PDB 7L1D) were obtained from VMD following a superposition of the Cα atoms of the residues 1-180 of the HLA-A3 peptide binding grooves and selection of all pTrp6 atoms. The van der Waals (vdW) overlap between pTrp6 and pHis2/pLeu2 was computed by PyMOL's 'overlap' command, with overlap for a pair of atoms determined by the difference between the atomic distance between them and the sum of their respective vdW radii. Peptide φ/ψ torsions from enforced rotation trajectories were measured using the MDAnalysis Python library[103]. Allowances for side chain torsion angles were determined by first collecting the φ/ψ values for non-terminal peptide residues and plotting the associated Ramachandran profiles. Frames containing instances outside of allowable regions were then counted for peptide amino acids. Allowances were set by the Ramachandran reference data internal to the MDAnalysis library (Rama_ref), which defines allowed and generously allowed regions with boundaries containing 90% and 99% of the reference measurements[104].

## Fluorescence anisotropy

For fluorescence anisotropy measurements for peptide dissociation, in both the WT and neoantigen peptides, pGly5 was substituted with a 5-carboxyfluorescein-modified lysine. Experiments were performed on a Beacon 2000 fluorescence polarization instrument at 4 °C and 25 °C by mixing 100 nM fluorescein-labeled peptide/HLA-A3 complexes with an excess of 100 μM unlabeled peptide in 20 mM NaH$_2$PO$_4$, 75 mM NaCl, pH 7.4. The excitation wavelength was 488 nm and polarization was detected at 535 nm. Changes in anisotropy were recorded as a function of time. Dissociation kinetics of peptides were determined by fitting the anisotropy curve to a single or biphasic dissociation function in OriginPro 2024.

## Nuclear magnetic resonance

$^{19}$F NMR spectra were recorded using a Bruker 600 MHz NMR instrument with a QCI cryogenic probe at NMRFAM, University of Wisconsin-Madison. All proteins were solubilized in 10 mM HEPES, 150 mM NaCl, pH 7.4 with 10% D$_2$O (Cambridge Isotope Laboratories DLM-4-25). Protein concentrations were 0.5 mM and 0.3 mM for the neoantigen/HLA-A3 and WT peptide/HLA-A3 complexes, respectively. For the TCR3-neoantigen/HLA-A3 complex, the sample was prepared by mixing neoantigen/HLA-A3 and TCR3 to final concentrations of 0.5 mM and 2.25 mM, respectively.

NMR probe temperature was maintained at a calibrated value of 5.2 °C. The large molecular weight of the peptide-protein complexes and the fast transverse relaxation of the $^{19}$F nuclear spin resulted in low detection sensitivity of the fluorine signals. To improve the signal/noise ratio, we compared $^{19}$F 1D spectra recorded with and without proton decoupling. Decoupling caused a noticeable reduction of the signal intensity of the $^{19}$F nuclear spin in the context of the complexes; therefore, all spectra were collected un-decoupled to maximize sensitivity. The reference frequency of the $^{19}$F channel was set through internal routines of the Topspin acquisition software after the spectrometer was locked on D$_2$O. The recycle delay was optimized to find the best compromise between the longitudinal relaxation and the

repetition rate for signal averaging. The optimal values of D1 = 0.4 s and AQ = 0.59 s were used in all experiments. One-dimensional fluorine experiments were recorded with the pulse program zgflqn.F from the standard Bruker library with TD1 = 134,078 points and a spectral width (SW) of 200 ppm. The spectra were acquired as 4 h blocks (NS = 13,500) interleaved with proton 1D recordings to monitor the sample stability, as well as re-tuning followed by re-shimming between the blocks for improved spectrometer performance. Free induction decays were processed in Mestrelab MNova NMR with an exponential window function, Fourier-transformed, and baseline-corrected prior to fitting with Lorentzian line shapes. The 4 h data blocks were checked individually for a possible frequency drift (which was not detected) and added to obtain combined datasets with improved signal/noise ratio. Multi-block acquisition mode allowed us to continue acquiring data blocks until an acceptable sensitivity in the combined dataset was achieved. The required total acquisition times of the 1D experiments were 12 h for the neoantigen complex, 33 h for the WT peptide complex, and 75 h for the TCR3-neoantigen/HLA-A3 complex to compensate for progressive broadening of the resonances due to conformational exchange (in the WT peptide complex) and increased molecular weight (in the TCR complex). For the data in Supplementary Fig. 9D, E total acquisition times were 20 min for both complexes. Line shape deconvolution was performed using MNova.

Conformational exchange saturation transfer (CEST) between $^{19}$F peptide signals in different protein environments was detected using the pulse program for the one-dimensional experiments modified to include a low-power presaturation pulse during the recycle delay[105]. Presaturation of a selected resonance was achieved by applying 10–40 μW irradiation on a fluorine channel at a desired ppm position for the duration of D1. The bandwidth of the presaturation pulse was directly measured using a free peptide sample by applying the pre-saturation pulse at different ppm positions. The direct effect of the pre-saturation pulse was found to be vanishing outside of +/−0.5 ppm from the presaturation frequency. The reference experiments were recorded with the presaturation pulse applied at −130 ppm (away from the peptide resonances) and were utilized to assess the magnitude of the baseline distortion artifacts caused by the presaturation. The CEST series was recorded as separate one-dimensional experiments with specific presaturation frequencies. The total acquisition time for each individual dataset was 2 h (NS = 7168).

Two-dimensional EXSY experiments were performed with the standard Bruker NOESY pulse program noesyphflqn modified to use the fluorine channel of the QCI probe[106] and a mixing time of 50 ms. The neoantigen/HLA-A3 dataset was collected with NS = 1024, TD1 = 32 and SW = 10.5 ppm for 10 h. The WT peptide/HLA-A3 dataset had a weaker signal, therefore, the SW and TD1 were modified to increase sensitivity at the expense of resolution: NS was set to 5120 with TD1 = 16 and SW = 6 ppm. Data were collected in four six-hour blocks (NS = 1280) that were combined after processing. To observe possibly slower exchange dynamics for the non-exchanging sharper resonance in the WT peptide/HLA-A3 sample, we collected an additional EXSY dataset with a 200 ms mixing time; no cross-peaks were observed, confirming an absence of detectable exchange dynamics between the sharp peak and other resonances of WT peptide/HLA-A3 sample.

## Statistics

Quantitative fitting of binding and kinetic dissociation data was performed in OriginPro 2024. Quantitative fitting of NMR lineshapes was performed in Mestrelab MNova NMR. T tests were performed with GraphPad Quickcalcs (https://www.graphpad.com/quickcalcs/ttest1.cfm).

## Reporting summary

Further information on research design is available in the Nature Portfolio Reporting Summary linked to this article.

## Data availability

All data are included in the Supplementary Information or available from the authors or are available from the Zenodo repository under record number 14392129. Crystallographic data are available from the PDB under accession codes 8VCL (neoantigen/HLA-A3; https://www.rcsb.org/structure/8VCL) and 9ASG (Bta-substituted neoantigen/HLA-A3; https://www.rcsb.org/structure/9ASG). NMR data are also available from the Biological Magnetic Resonance Bank under accession code BMRbig105 (https://bmrbig.org/released/bmrbig105). The raw numbers for charts and graphs are available in the Source Data file whenever possible. Source data are provided with this paper.

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

## Acknowledgements

This research was supported by National Institutes of Health (NIH) grants R35GM118116 to B.M.B. and R37CA259177 to C.A.K. C.A.K. further acknowledges the support of NIH grants R01CA269733, R01CA286507, P30CA008748, the Parker Institute for Cancer Immunotherapy, the Metropoulos Family Foundation, the Damon Runyon Cancer Research Foundation (CI-96-18), the Breast Cancer Alliance, the Manhasset Women's Coalition Against Breast Cancer, the Cancer Research Institute (CRI3176), and a sponsored research agreement with Intima Bioscience. This work is based in part upon research conducted at the NE-CAT beamlines at the Advanced Photon Source, which are funded by the National Institute of General Medical Sciences from the National Institutes of Health (P30 GM124165), as well as the GM/CA beamlines which have been funded by the National Cancer Institute (ACB-12002) and the National Institute of General Medical Sciences (AGM-12006, P30 GM138396). This research used resources of the Advanced Photon Source, a U.S. Department of Energy (DOE) Office of Science User Facility operated for the DOE Office of Science by Argonne National Laboratory under Contract No. DE-AC02-06CH11357. This study also made use of the National Magnetic Resonance Facility at Madison, where NMR equipment, helium recovery equipment, and computers were purchased with funds from the University of Wisconsin-Madison and the NIH (awards P41GM136463, R24GM141526, P41GM103399, S10RR023438, S10RR025062, and S10RR029220). The use of SPR was supported partially by NIH grant S10OD028553. The authors are grateful to Dr. Marco Tonelli for providing pulse programs for NMR experiments and assisting with access to the NMR spectrometer. We acknowledge the CCP4/APS School in Macromolecular Crystallography for assistance in X-ray data collection and structure solution.

## Author contributions

J.M.: crystallography; binding, stability, kinetic, and NMR experiments; traditional MD simulations; WEMD simulations and analyses; C.A.: traditional MD simulations and analyses; C.B.: SMD simulations and analyses; S.S.C.: T cell functional experiments and analyses; T.R.: WEMD simulations and analyses; G.P.: crystallography; B.E.: binding and stability experiments; W.M.: T cell functional experiments; S.A.C.: guidance on WEMD simulation implementation and interpretation; E.K.: NMR experiments and analyses; C.K.: study design, data interpretation, funding; B.B.: study design, data interpretation, funding; J.M., C.A., C.B., S.S.C., T.R., S.A.C., E.K., C.K., B.B.: initial draft; All authors: final draft.

## Competing interests

C.A.K. and S.S.C. are inventors of patents related to the T cell receptor (TCR) sequences featured in this manuscript and are recipients of licensing revenue from Intima Bioscience shared according to Memorial Sloan Kettering Cancer Center (MSKCC) institutional policies. C.A.K. has consulted for or is on the scientific advisory boards for Achilles Therapeutics, Affini-T Therapeutics, Aleta BioTherapeutics, Bellicum

Pharmaceuticals, Bristol Myers Squibb, Catamaran Bio, Cell Design Labs, Decheng Capital, G1 Therapeutics, Klus Pharma, Obsidian Therapeutics, PACT Pharma, Roche/Genentech, Royalty Pharma, and T-knife, and is a scientific co-founder and equity holder in Affini-T Therapeutics. S.S.C. is a scientific advisor and equity holder in Affini-T Therapeutics. B.M.B. is an inventor on patents relating to differences between mutant and self in identifying immunogenic neoantigens has consulted for or received funding from Merck, Pfizer, Eureka Therapeutics, and EnaraBio, and is on the scientific advisory board of T-cure Bioscience. The other authors declare no competing interests.
