## [Transparent Peer Review file · Nature Communications]

Dynamic allostery in the peptide/MHC complex enables TCR neoantigen selectivity

Corresponding Author: Professor Brian Baker

Version 0:

Reviewer comments:

Reviewer #1

(Remarks to the Author)

Ma et al present the paper entitled 'Dynamic allostery in the peptide/MHC complex enables TCR neoantigen selectivity'. Neoantigens derived from single amino acid substitutions present an extreme challenge to the immune system to discriminate between 'self' and 'non-self'. The location of the substitution can determine the mechanism of neoantigen discrimination. When substitutions occur at anchor residues, the newly generated MHC bound peptide is entirely 'non-self' and maybe easier to detect. For substitutions within existing 'self' peptides, TCRs can focus on the site of the substitution. However, other structural mechanisms for TCR detection of neoantigens are being uncovered including for when only minor conformational changes are observed between WT and neoantigen peptide derived from static crystal structures. Computational prediction of immunogenicity is an important goal for the field and understanding the structural basis for neoantigen discrimination can contribute to such computational models.

Here, Ma et al. describe a novel allosteric mechanism for TCR detection of neoantigens. The same group previously identified multiple TCRs specific for the neoantigen ALHGGWTTK derived from a common mutation in PI3KA. The main conclusion is that detection of this neoantigen requires a dramatic allosteric change in the orientation of a p6 Trp residue that is conserved between WT and neoantigen peptides. This p6Trp orientation was required for TCR recognition and was only observed in the TCR complexed with neoantigen bound MHC, while the same neoantigen peptide in the TCR free structure adopted a very similar structure to the WT peptide. Using molecular dynamics and NMR spectroscopy, the authors determined that the p6TRP must flip from facing the alpha1 helix to the alpha 2 helix only by passing underneath the peptide backbone and not over the top. This mechanism relies on the existence of a contiguous cavity between the peptide and MHC binding groove with sufficient space to allow p6Trp to rotate from facing the α 1 helix to the α 2 helix.

The work presented in this paper was carried out to a very high standard and represents an important conceptual advance for the field. I particularly enjoyed the use of the Trp analog Bta to support their model. Understanding how TCRs detect neoantigens is important for building computational tools that predict immunogenicity. Furthermore, structural understanding of TCR recognition of neoantigens is important to develop a molecular basis for understanding cross-reactivity. This is especially important for TCRs being developed for clinical use, such as the TCRs examined in this study. I have a few comments.

1) The allosteric model presented requires a cavity beneath peptide that would accommodate the p6 Trp. I appreciate the analysis of cavity sizes in other A3 supertype structures but I was wondering if the authors could describe this specific cavity in more detail – what residues line the cavity and what role do they have in the static HLA-A3 structures? Further, is it possible to occupy this cavity by making substitutions that would otherwise not impact peptide-MHC contacts? These mutants would likely still bind the neoantigen but prevent the p6Trp flip?

2) The MD simulations and NMR data are extremely important for the conclusion of the paper. However, the description of these data is to me, a little too long with some methods described in detail. I fear this may dissuade many readers from uncovering how the authors reached the conclusion of the p6Trp flip. I would encourage the authors to edit these sections for clarity and brevity, with goal of inviting many non-structural biologists to read their work.

- 3) While not the focus of the study – could the authors comment on the success/failure of computational based structure predictions (eg AlphaFold) in identifying the correct Trp6 orientation defined by their work? Are these allosteric changes identified or are non-computational methods still needed for discoveries such as theirs?
- 4) Figure 1b in the text refers to molar peptide concentrations but the figure displays mass concentrations – please correct.
- 5) The authors use capitalized italics for the PI3K neoantigen, which is typically reserved for gene names.

Reviewer #2

(Remarks to the Author)

In this manuscript, Ma et al. extend their previous study of TCR recognition of a public neoantigen derived from the PIK3CA oncoprotein. The authors first provide evidence from MD simulations that the amino acid at anchor position 2 of PIK3CA influences the motions of the peptide in the HLA-A3 binding groove, such that the more rigid neoantigen, with Leu rather than His at position 2, is better able to transition Trp at position 6 to a flipped conformation required for TCR binding. Experimental support for MD simulations is provided by NMR spectroscopy showing that the position 2 residue alters the motions of Trp at position 6. The authors thereby propose dynamic allostery as the mechanism responsible for TCR selectivity for mutant PIK3CA.

Although this is a reasonable and very interesting conclusion, it should be noted that the authors previously reported that the melting temperature (i.e. thermal stability) of the wild-type PIK3CA–HLA-A3 complex (37 oC) is considerably lower than that of the mutant PIK3CA–HLA-A3 complex (54 oC) (ref. 15). In addition, the half-life of the wild-type PIK3CA–HLA-A3 complex (0.08 h) is much shorter than that of the mutant PIK3CA–HLA-A3 complex (5.5 h). Why are these very large difference not sufficient to explain the inability of wild-type PIK3CA to stimulate T cells that recognize mutant PIK3CA? The authors need to justify their rationale for seeking more complicated explanations than simply stability differences arising from a suboptimal anchor residue (His) at position 2 in wild-type PIK3CA that is mutated to a preferred residue (Leu) in the neoantigen. This, after all, was the explanation the authors gave in their original publication.

Reviewer #3

(Remarks to the Author)

In this manuscript Ma et al, investigate the interaction of TCRs with a neoantigen derived from PIK3CA bound to HLA-A*03:01 to uncover determinants of T cell specificity beyond static structural models derived from X-ray crystallographic analysis. The study utilizes TCRs generated in a prior study from healthy donors challenged with the PIK3CA epitopes via in vitro expansion (ref 15). The authors examine the previously generated structures of these TCR-pMHC complexes by herein utilizing computational molecular dynamics (MD) and 19F-NMR dynamics studies of the pMHC in the absence of the TCR. They identify elements of conformational exchange which they deem critical for ligand recognition by the cognate TCR. The authors suggest a mechanism for the allosteric rearrangement of the H2L variant but not the WT peptide, bound to HLA-A*03:01, that determines the specificity of the TCRs.

The paper represents a serious examination of the subject matter through multiple structure-based analytic techniques which is a major strength of the current study. Such multidimensionality in approach deserves recognition as the task is challenging. The methods are sound and sufficient details provided to allow others to replicate the work. However, a key deficiency is the absence of MD studies on the TCR-pMHC complex itself, for reasons noted below. Further, some points must be addressed or clarified prior to consideration for publication. If modified aptly, the manuscript should be worthy of publication in the Journal.

Major:

1. On p5 para 1 the authors state, “ Even though less potency is expected from the WT peptide given its weaker binding to HLA-A3, in other instances where peptides modified at the first primary anchor have been examined, weaker binding peptides have still elicited quantifiable functional responses (16, 23, 24).” This seems an oversimplification, as the t1/2 at 37C is 100 times lower for the WT. The MART-1 comparison is less apt as the t1/2 differential is only 4 fold (ref 18). The lack of detection of the WT peptide by MS as determined in ref 15 fig3 seems to show that the correct explanation is lack of WT presentation. Although there are some differences in methodology in measurements quoted above, it would be helpful if the authors are more explicit as to why the rationale for presentation of the WT epitope has changed since the prior publication? See also point 3 below.
2. p6W does not directly contact TCR4 in the crystal structure, yet the pH3 does, forming contacts with both CDR3a and CDR1a. The p5G does form contact with CDR3a, but it is unclear how the conformational change is affecting the direct interactions using the current figures. It would be very helpful to have a stick view of an overlay to show the differences more readily between peptide conformation of unligated versus bound to TCR4 and TCR3 in fig 2a instead of the ribbon peptide view currently provided. One could simply add two panels to the current figure to illustrate this point more clearly, providing a more incisive view.
3. The implication is that in the unbound pMHC p6W is solvent exposed and would be readily available for TCR binding. Is this true? How could such contact facilitate structural rearrangement of p6W? See query #7 below as well since it is directly related to this query and needs to be addressed.
4. The viable alternate hypotheses functionally are that (a) as suggested in ref 15, WT peptide is never presented effectively on the cell surface and that there are potentially TCRs that interact both with WT unflipped p6W and the p2L variant in both flipped and unflipped p6W states, with cross-reactivity possible particularly for beta-dominant TCRs, or (b) WT peptide is

presented effectively as posited in this manuscript and all WT-specific TCRs are negatively selected during thymocyte development, leaving only TCRs that interact with the flipped p6W and thus only recognizing mutant pMHC. Is (was) it possible to isolate a WT-specific TCR from healthy volunteers? Was such an effort made? Is it likely that TCRs could recognize the unflipped neoantigen-MHC? Can this information be readily gleaned from MD? Is there any further evidence that the recognition mode of TCR3 and TCR4 is typical of TCRs found in humans that recognize this neoepitope? Could this be commented on, if so.

5. In the discussion about “structurally silent” recognition elements, the authors reference the A6 TCR system recognizing a cognate HTLV1 Tax peptide and its closely related variants which yielded highly disparate functional T-cell activation results. While this seems germane to the present manuscript, as referenced, it is also appropriate to reference and discuss a recently published paper demonstrating dynamic differences between the TCR-pMHC in this system as studied with and without force (Chang-Gonzalez et al. eLife 2024; 13:e91881. DOI: <https://doi.org/10.7554/eLife.91881>). Relevantly, the application of force is also a variable not accounted for in static structural models on which Baker was a coauthor years ago. The eLife MD study with and without force application not only demonstrates dynamic allostery in the TCR system but offers the rational basis for explaining functional differences in structurally highly similar pMHC ligands interacting with the same TCR. Hence the omission of this reference and discussion in the current context is glaring.

6. The historical term “structurally silent” is, in fact, a misnomer and should be explicitly called out in the abstract, introduction and/or discussion. As this manuscript aptly shows, the differences are not silent except by virtue of the technique used by investigators to interrogate the structure, in this case being restricted to X-ray crystallography.

7. Molecular dynamic considerations: Using unbiased molecular dynamics (MD) simulation as well as biased simulations, they argue that the motion of H2 in the WT peptide results in a high barrier for turning W6 towards the alpha2 helix of MHC. While the simulation and analyses performed are overall well-done, there are some concerns as follows:

The alpha2-facing orientation of W6 was observed in x-ray structures of the system with a bound TCR (PDB 7L1D & 7RRG), whereas all the simulations in this manuscript were performed in the absence of TCR. Without considering TCR, the of important the proposed mechanism is uncertain. Pointedly, the interaction with the TCR may well change the energy landscape associated with flipping of W6, and the low affinity for TCR of the WT system could be due to a different mechanism, e.g., simply due to a higher mobility of the N-terminal part of the WT peptide (Fig 4A) that cannot stabilize the bound state, and/or the less bulky WT H2 (compared to L2 of the mutant) makes the peptide slightly closer to the MHC floor (as they observed), in which case the motion of H2 per se may not be as important. Ignoring the interaction with TCR is the biggest shortcoming of this current MD study. This should be rectified.

By restricting the scope of the current manuscript to the pMHC system important insights can be missed. Could the interaction between TCR and pMHC differentially augment W6 flipping or otherwise alter recognition of WT vs mut pMHC with or without force? Does this relate to differential recognition in the context of highly related HLA molecules (HLA-A*03:01 serves to mediate mutant T-cell recognition whereas HLA -A*03:02 and -A* 11:01 do not) noted in their prior Nature Medicine 2022 study? How do the authors think about patients whose T cell recognize the neoepitope, yet CTL effector function may not be adequate to mediate tumor protection since a clinical tumor is evident? Is efficiency inadequate because of requisite structural rearrangements? This last question is not meant to be answered but rather food for thought.

Based on difference in the motion of the residue at position 2 affecting the behavior of W6 along the peptide, the authors use the term ‘dynamic allostery’. It is unclear whether use of the term is suitable here. The WT peptide has H2 and the mutant has L2, with the latter peptide residue being bulkier and more hydrophobic. As mentioned above, is it the motion of H2 that prevents W6 flipping, or the conformation of the peptide that differ slightly from L2 that is responsible for the difference? How are these motions impacted by TCR ligation?

Minor:

1. Please provide reference for CD107a (LAMP-1) assay in the methods section. Why this vs. IL2 or CD69? Have the authors found this technique more sensitive or otherwise advantageous over the others or just comparable as a point of information.
2. On p6 para 4 , “...feature of PIK3CA neoantigen recognition by multiple TCRs, ...” , should the word “multiple” be changed to “two” as there are only two defined structures?
3. Is there a volumetric measurement available for the 3D conformational sampling represented by occupied voxels in Figs 4d-e and Fig 5a? It might be helpful to the reader to have some quantitation?
4. Regarding the ED Fig 6e -124.92 ppm “degradation” peak in the 19F spectra, is it likely that this is free peptide non-specifically associated with unfolded HLA?
5. Please provide names of Bruker pulse sequences used for CEST and EXSY with references. Please provide SW, number of increments and scans for the CEST. Also, were spectra processed with Topspin? Was the unsaturated reference with no pulse or off-resonance pulse?
6. Histidine protonation states chosen for simulation are not mentioned in the manuscript. This is especially important for H2 and H3 of the peptide, which could impact dynamics of the peptide.

7. p9 “rotation of Tyr99 was slightly (~1ns) preceded by rotation of pHis2”:
Without more quantitative analysis, this statement cannot be concluded from fig 5C

Version 1:

Reviewer comments:

Reviewer #1

(Remarks to the Author)
No further comments.

Reviewer #2

(Remarks to the Author)
The authors have responded satisfactorily to the previous critiques,

Reviewer #4

(Remarks to the Author)
I did not review this manuscript initially, so have focused most of my attention on the response to reviewers and whether the authors have revised the manuscript appropriately to the reviewers comments. I have read through the reviewers' comments (all were quite positive and constructive). I've also read through the authors' response to these critiques and evaluated how they revised the manuscript accordingly. I found the revision thorough, thoughtful and the manuscript quite outstanding. I do not see any reason why this manuscript should not be accepted; it will be of broad interest to the Nat Coms readership.

We thank the reviewers, editor, and journal staff for their efforts on our manuscript. We are grateful that all three reviewers found significant strengths, but we also acknowledge the questions and concerns that arose during review. We are thankful for the opportunity to submit and have addressed each question via the comments below and in changes to the manuscript. In responding we have collected new X-ray, NMR, and simulation data, and revised the presentation and discussion of the data. Our responses and changes to the manuscript are described below in blue text. Significant changes or additions to the manuscript are highlighted in red in the manuscript file.

Reviewer #1

“The work presented in this paper was carried out to a very high standard and represents an important conceptual advance for the field. I particularly enjoyed the use of the Trp analog Bta to support their model. Understanding how TCRs detect neoantigens is important for building computational tools that predict immunogenicity. Furthermore, structural understanding of TCR recognition of neoantigens is important to develop a molecular basis for understanding cross-reactivity. This is especially important for TCRs being developed for clinical use, such as the TCRs examined in this study.”

We appreciate the reviewer’s favorable comments and their assessment of how we used non-natural amino acids to probe the system.

“1) The allosteric model presented requires a cavity beneath peptide that would accommodate the p6 Trp. I appreciate the analysis of cavity sizes in other A3 supertype structures but I was wondering if the authors could describe this specific cavity in more detail – what residues line the cavity and what role do they have in the static HLA-A3 structures? Further, is it possible to occupy this cavity by making substitutions that would otherwise not impact peptide-MHC contacts? These mutants would likely still bind the neoantigen but prevent the p6Trp flip?”

The prevalence of large cavities between the base of the groove and the peptide initially surprised us, but in hindsight, this is sensible given the architecture of class I molecules, particularly the need for peptides to bulge from the groove for TCR surveillance and recognition. Nonetheless, the distribution of cavity sizes varies considerably based on individual pMHC structure, especially if peptides have bulky side chains that are oriented “down” towards the base of the groove.

In responding to this comment, we added a figure that shows the residues that line the cavity in the neoantigen/HLA-A3 complex (**new Supplemental Fig. 3**). We did consider mutating these to impact the ability of the peptide to flip, but this proved challenging – in some cases the amino acids are already large and bulky or packed in the base of the groove (e.g., W133, W147, R114, L156). In other cases, the residues also contact the peptide or one of the TCRs in the bound state or the peptide (e.g., N66, D77, E152), making mutations here difficult to interpret. Thus, although we agree it would be an interesting experiment, there isn’t an obvious target where we could make a mutation to “fill up” the cavity.

We did try mutations in the base of the groove that would alter the propensity for the TCR to flip, which would also alter the volume of the cavity. As Y99 is implicated in the “gate” that permits the neoAg to flip but restrains the WT peptide, we changed Y99 to phenylalanine, serine, and threonine. We selected these to test the hypothesis that making position 99 smaller would open the gate, allowing the WT peptide to flip, and also because Phe and Thr are seen in this position in other HLA-A alleles. Unfortunately, these mutant HLA-A3 molecules were very unstable – we could produce a small amount of recombinant complex with the neoAg, but almost none with the WT protein. DSF data for the Y99F mutant with the neoAg is shown in the figure to the right. The instability of these complexes precluded further analyses, and given the additional complexities noted above, we did not pursue further mutations along the cavity wall.

DSF data showing the thermal stability of the neoAg wild type HLA-A3, as well as a mutant with Tyr99 changed to Phe. The mutation dramatically destabilizes the complex.

“2) The MD simulations and NMR data are extremely important for the conclusion of the paper. However, the description of these data is to me, a little too long with some methods described in detail. I fear this may dissuade many readers from uncovering how the authors reached the conclusion of the p6Trp flip. I would encourage the authors to edit these sections for clarity and brevity, with goal of inviting many non-structural biologists to read their work.”

We appreciate this point, particularly for the WESTPA and SMD simulations, which are more specialized than the traditional MD simulations. In responding, we went through these sections carefully and tried to remove some material, while condensing other sections. We note that in response to Reviewer 3, additional simulations for the complexes were added, and additional simulations were also added to meet the requirements of the *Nature Communications* reporting form for MD simulations (**new Supplemental Figs. 2, 4, and 5**). We are open to additional simplifications if the reviewer has further suggestions.

“3) While not the focus of the study – could the authors comment on the success/failure of computational based structure predictions (eg AlphaFold) in identifying the correct Trp6 orientation defined by their work? Are these allosteric changes identified or are non-computational methods still needed for discoveries such as theirs?”

We co-developed TFold, currently the most accurate AlphaFold-based approach for modeling pMHC complexes (and which also outperforms legacy-based modeling such as those based on Rosetta and MODELLER) (Mikhaylov *et al.*, *Structure* 2024). Using TFold for the neoantigen and WT pMHC complexes, when compared to the crystal structures we obtain peptide full atom RMSDs of 2.1 Å (WT) and 1.9 Å (neo). In both models, there is a misalignment of the backbone in the N-terminal halves of the peptides as well as the crucial position 6 side chain, which the AI wants to put down towards the base of the groove, as if it wants to flip (*figure below, left*). We note that the confidence scores for both models are below the threshold for a “good” model, indicating that AI-based modeling still requires improvements.

We also used TCRmodel2, a version of AlphaFold2 developed specifically for TCR complexes (PMID 37140040), as well as AlphaFold3 to model the structure of TCR4 with the neoAg/HLA-A3 complex. Neither performed particularly well – the AF3 model did move the peptide under the backbone, resulting in a partial flip. TCRmodel2 did not flip the peptide at all. Both methods performed poorly with the CDR3 loops, mispositioning CDR3 α and completely mismodeling the extended CDR3 β loop (*figure below, right*).

Comparison of the AI-based model of the neoAg in the HLA-A3 groove to that of the crystallographic structure. The model misplaces the N-terminal half of the backbone and the p6Trp sidechain.

Comparison of the AI-based models of TCR4-neoAg/HLA-A3 complexes. The AF3 model does initiate a peptide flip, while TCRmodel2 does not. Both AI models perform poorly with the CDR3 loops.

We agree with the reviewer that these are interesting results indicating both the promise and limitations of AI based modeling, but we also agree that this is outside the scope of the paper. Given the paper’s

current length and complexity, we hope the reviewer will agree that including this in the paper would be distracting, especially given that AI-based pMHC modeling has been addressed (TFold manuscript noted above), as well as recent work from Pierce and Mariuzza showing that AI-based TCR-pMHC modeling also requires further development (PMID 38826362).

“4) Figure 1b in the text refers to molar peptide concentrations but the figure displays mass concentrations – please correct.”

We thank the reviewer for catching this. This is fixed in the revised version of the figure.

“5) The authors use capitalized italics for the PI3K neoantigen, which is typically reserved for gene names.”

The reviewer is correct - *PIK3CA* refers to the gene. The protein is referred to as PI3K α . We could make an argument to use either and went with the former in our prior manuscript, rationalizing that the neoAg originates from the genetic sequence. After reflecting on this though, here we eliminated the use of *PIK3CA* except where explicitly referring to the gene and replaced it with PI3K α . To make the manuscript a little more accessible, we also tried to reduce the number of times we use PI3K α as we also found this distracting on a re-read.

Reviewer #2

“In this manuscript, Ma et al. extend their previous study of TCR recognition of a public neoantigen derived from the PIK3CA oncoprotein. The authors first provide evidence from MD simulations that the amino acid at anchor position 2 of PIK3CA influences the motions of the peptide in the HLA-A3 binding groove, such that the more rigid neoantigen, with Leu rather than His at position 2, is better able to transition Trp at position 6 to a flipped conformation required for TCR binding. Experimental support for MD simulations is provided by NMR spectroscopy showing that the position 2 residue alters the motions of Trp at position 6. The authors thereby propose dynamic allostery as the mechanism responsible for TCR selectivity for mutant PIK3CA.”

This is essentially the story, although we would add that the alterations of the motions in the peptide and HLA protein are what alter the propensity of the tryptophan to flip.

“Although this is a reasonable and very interesting conclusion, it should be noted that the authors previously reported that the melting temperature (i.e. thermal stability) of the wild-type PIK3CA–HLA-A3 complex (37 oC) is considerably lower than that of the mutant PIK3CA–HLA-A3 complex (54 oC) (ref. 15). In addition, the half-life of the wild-type PIK3CA–HLA-A3 complex (0.08 h) is much shorter than that of the mutant PIK3CA–HLA-A3 complex (5.5 h). Why are these very large difference not sufficient to explain the inability of wild-type PIK3CA to stimulate T cells that recognize mutant PIK3CA? The authors need to justify their rationale for seeking more complicated explanations than simply stability differences arising from a suboptimal anchor residue (His) at position 2 in wild-type PIK3CA that is mutated to a preferred residue (Leu) in the neoantigen. This, after all, was the explanation the authors gave in their original publication.”

This is a fair point. At the outset, we emphasize that we do not address *in vivo* activity, but the concordance between the lack of any measurable *in vitro* response and the inability to detect TCR binding, and how motional properties contribute to these – i.e., this is a story of TCR specificity in molecular recognition.

There are four major points to our interpretation of the data. **First**, we were unable to detect functional recognition of the WT peptide in multiple *in vitro* functional assays, even at very high peptide concentrations. This was surprising to us given the nearly identical structures of the neoAg and WT peptide.

To further drive this point home, in the revised manuscript we now include another structure of the neoAg/HLA-A3 complex (**updated Fig. 1A**), which is even more similar to the WT (the first structure shows a slight variance in the center of the backbone, which we noted a TCR could possibly sense, but this is lacking in this second structure). We attribute these variances is as evidence of the dynamics we see by simulation and NMR (mentioned in second paragraph of the results on **page 5**).

Second, in other instances of tumor antigens where suboptimal anchors are present, although physiological *in vivo* responses may not always be clear, *in vitro* responses *have* been detected. The textbook example for this is the MART-1 tumor antigen (nonameric form), which is poorly immunogenic with numerous clones, but nonetheless leads to detectable response in *in vitro* assays with many. In our case, the affinity of the WT PI3K α peptide to HLA-A3 is the same as that of the MART-1 tumor antigen to HLA-A2, yet with multiple T cell clones and multiple readouts, we have been unable to detect *in vitro* responses to the α peptide. Notably, when peptide dissociation kinetics are compared using the same assay, the half-life of the WT peptide/HLA-A3 complex is more than twice as long as that of MART-1/HLA-A2 (see refs. 10 and 17 and the top of **page 5** in the revised manuscript, as well as the response to point 1 of Reviewer 3 below).

Third, we note our work in other systems where anchor mutations alter TCR recognition. The clearest example is our prior work with gp100-specific TCRs, where we showed that, depending on the TCR, substitutions at the position 2 anchor that improve peptide binding can *strengthen or weaken* TCR affinity and *in vitro* functional potency, again with no apparent changes in pMHC crystallographic structures (Smith et al., *PNAS* 2024, ref. 6 in the revised manuscript). This work is mentioned as part of the rationale for our work on **page 3** in the Introduction.

Fourth, and perhaps most crucially, we are unable to detect binding of the TCRs to the WT complex, *despite the sample being fully assembled and binding competent as shown by our s3-4 scTv positive control (Fig. 1D)*. This latter experiment is crucial for our interpretation, as it shows that *at the molecular level*, the two TCRs we study clearly distinguish between the two peptides. This is mentioned at the top of the last paragraph on **page 5** and at the end of the second paragraph on **page 6** and is fully consistent with the functional data showing no recognition.

As the reviewer implies, we cannot with confidence say what is happening *in vivo* in this case, but we note that is not the point – we are emphasizing the ability of TCRs to distinguish between the two peptides, or equivalently, show how anchor modification renders the two distinct despite the overlapping crystallographic structures.

We also have other data that supports our interpretation, which is shown below. When examining other amino acids at position 2 of the peptide, we found that changing the suboptimal histidine to valine did not yield a peptide that could be functionally recognized, ***despite the substitution making the peptide bind better to HLA-A3, and despite not altering the peptide's structure in the binding groove***. Although we have not fully investigated this variant (we refer to it as AVH, with the bold V representing the p2 His \square Val substitution), we did use the structure to complete similar MD simulations on the pMHC complex. The data suggest that with valine at position 2, the peptide retains the motional properties of the WT peptide. This data is shown in the figure below and includes the crystallographic structure, SPR data examining TCR binding, peptide binding data using DSF, a functional titration, and the results of the MD simulations.

This result further demonstrates the allosteric coupling between the p2 amino acid and what happens in the middle of the peptide. It is consistent with our work in the gp100 system where we show how different p2 anchor residues impact TCR binding independently of peptide binding (ref. 6 in the revised manuscript).

We have not included this data in the revised manuscript because a) we believe our point is already made via points 1-4 above (with revisions to the manuscript as noted), and b) the manuscript is already long and complex, especially with the new data added in response to the review (see response to Reviewer 3 below). We plan to use this work with the AVH peptide as the basis for a follow-up story.

(Redacted)

(Redacted)

We do agree that the lack of a detectable immune response towards the WT peptide is likely driven in part by the lower stability of the WT pMHC complex – work from years ago on gp100 and MART-1 showed that better binding peptides can be more immunogenic *in vitro*, and we make this point on **pages 4-5** of the revised manuscript (and cite refs. 16, 23, and 24). However, our work shows that primary anchor modification also alters the ability of TCRs to bind the peptide/MHC complex, just as prior work was shown to do so with gp100, MART-1, and NY-ESO-1. How this plays out *in vivo* we cannot directly say – it is certainly tempting to conclude that weaker peptide binding plays a role *in vivo* and we would not argue against that. We are cautioned though by other work that has shown that T cells respond to neoantigens that bind exceptionally poorly to their MHC-I proteins, and in mouse models, very weak binding peptides that show no activity *in vitro* can actually drive anti-tumor immunity and protect animals from death when used as vaccines (refs. 71-73 in the revised manuscript).

We have made several changes to the manuscript to address these points. As noted above, we included a new structure that emphasizes the essential identical crystallographic structures of the neoAg and WT peptide/HLA-A3 complexes (**updated Fig. 1A**). We also made small changes to the Introduction around this point (changes to **pages 3 and 4**) emphasizing TCR recognition (as distinct from tumor eradication *in vivo*), our prior work with gp100-specific TCRs, and the MART-1 data. We also make this point again on **pages 5 and 6**, where we emphasize TCR recognition, the identical structures, and point out how the MART-1 peptide can be recognized functionally despite dissociating faster than the PI3K α neoAg. We take special care to point out the importance of the SPR experiment showing our inability to detect TCR binding, as well as the s3-4 positive control which shows this is not due to sample degradation from peptide dissociation (**page 6 and Fig. 1D**).

We also made changes to the Discussion, where we bring up *in vivo* biology (**page 18**). We explicitly point out that our results do not speak to *in vivo* immunogenicity and highlight the need for further work to address the interplay between TCR binding to pMHC and peptide binding to MHC in determining immunogenicity.

Lastly, to the reviewer's final point ("*This, after all, was the explanation the authors gave in their original publication*") - in our 2022 paper, although we did make note of the higher affinity the neoAg has for HLA-A3, we did not conclude this was the basis of immunogenicity. The neoAg was not identified by first predicting MHC binding affinity as is often done in neoAg prediction. Instead, we performed a functional screen to find TCRs that recognized targets making mutant *PIK3CA* and identified the neoantigen from there. We did point out the high specificity of the resulting TCRs, which include the ability to distinguish not only between the neoAg and its WT counterpart. When comparing the structures of the neoAg and WT peptide/HLA-A3 complexes, our comment was that "*The overall similarities of the two complexes suggest that structural differences alone likely cannot explain the immunogenic potential of the PIK3CA public NeoAg.*" The closest we came to making the reviewer's point was the statement in the results section "...that the formation of the *PIK3CA* public NeoAg is driven by the creation of a peptide sequence containing a preferred HLA-I anchor residue, resulting in a stable, high-affinity p/HLA-I complex with a prolonged *t*_{1/2}."

Reviewer #3

"The paper represents a serious examination of the subject matter through multiple structure-based analytic techniques which is a major strength of the current study. Such multidimensionality in approach deserves recognition as the task is challenging. The methods are sound and sufficient details provided to allow others to replicate the work. However, a key deficiency is the absence of MD studies on the TCR-pMHC complex itself, for reasons noted below. Further, some points must be addressed or clarified prior to consideration for publication. If modified aptly, the manuscript should be worthy of publication in the Journal."

We thank the reviewer for recognizing the depth and dimensionality of our work.

Major points:

"1. On p5 para 1 the authors state, "Even though less potency is expected from the WT peptide given its weaker binding to HLA-A3, in other instances where peptides modified at the first primary anchor have been

examined, weaker binding peptides have still elicited quantifiable functional responses (16, 23, 24).” This seems an oversimplification, as the $t_{1/2}$ at 37°C is 100 times lower for the WT. The MART-1 comparison is less apt as the $t_{1/2}$ differential is only 4 fold (ref 18). The lack of detection of the WT peptide by MS as determined in ref 15 fig3 seems to show that the correct explanation is lack of WT presentation. Although there are some differences in methodology in measurements quoted above, it would be helpful if the authors are more explicit as to why the rationale for presentation of the WT epitope has changed since the prior publication? See also point 3 below.”

This is similar to the concern of Reviewer 2, which we respond to in detail above, including showing new data that even making the peptide bind *better* to HLA-A3 still does not restore *in vitro* immunogenicity or detectable TCR binding, even though the peptide conformation stays the same (as we have seen in other systems; ref. 6).

We also appreciate the reviewer bringing up peptide dissociation kinetics, as we had not introduced these in the manuscript. The reviewer is correct in noting that the measurements in refs. 15 and ref. 17 are different techniques – the former uses DSF to measure protein unfolding/aggregation as a consequence of peptide dissociation, whereas the latter directly monitors peptide dissociation via fluorescence anisotropy. These physical processes are different, and as we wrote in ref. 17, the absolute values obtained by DSF differ from those obtained by other types of measurements, although the relative differences when peptides are compared are closer. Fortunately, we *have* measured MART-1 peptide dissociation from HLA-A2 using the exact same fluorescence anisotropy technique used for the PI3K α peptides (down to the same instrument). This data was published in JBC in 2011 (Insaideo et al., PMID 33468649, ref. 10 in the revised manuscript). The relevant data are in Fig. 6 of that paper, reproduced below (left).

At 37 °C (top panel), the MART-1 nonamer (AAG) dissociates from HLA-A2 with a k_{off} of 0.0055 s^{-1} . The value for dissociation of the PI3K α WT peptide from HLA-A3 at 37 °C is about two-fold *slower* at 0.0024 s^{-1} (half-lives of 126 seconds for MART-1, 289 seconds for PI3K α). The difference is evident if we plot both sets of data together (below right).

Fig. 6 from Insaideo et al., JBC 2011 (ref. 10).

In responding to this comment, we note that changes introduced in response to Reviewer 1 (including rewriting the quoted statement on **page 4**), but also emphasize our introduction of kinetics on **pages 4 and 5** of the revised manuscript, where we point out that the PI3K α WT peptide dissociates slower than the MART-1 peptide. We thank the reviewer for this suggestion.

“2. p6W does not directly contact TCR4 in the crystal structure, yet the pH3 does, forming contacts with both CDR3a and CDR1a. The p5G does form contact with CDR3a, but it is unclear how the conformational change is affecting the direct interactions using the current figures. It would be very helpful to have a stick view of an overlay to show the differences more readily between peptide conformation of unligated versus bound to TCR4 and TCR3 in fig 2a instead of the ribbon peptide view currently provided. One could simply add two panels to the current figure to illustrate this point more clearly, providing a more incisive view.”

This is a good suggestion. We added the requested figure as a new supplemental figure (**new Supplemental Fig. 1**). We note these changes were described in various ways in our prior manuscript, including summaries of all the ϕ/ψ bond changes, etc.

“3. The implication is that in the unbound pMHC p6W is solvent exposed and would be readily available for TCR binding. Is this true? How could such contact facilitate structural rearrangement of p6W? See query #7 below as well since it is directly related to this query and needs to be addressed.”

The side chain of p6W is solvent exposed as shown in **Fig. 3B** and mentioned in the first paragraph on **page 7**. As to why the peptide undergoes a conformational change, answering precisely why changes occur in proteins is always a challenge, but we addressed this in the Discussion (**page 18**). The interpretation most consistent with the data relates to changes in solvation requiring a hydrogen bond to the Trp, which we validated is crucial for TCR binding with the Bta experiment (**Figs. 3D-G, discussed on pages. 6-7**). We discuss this issue further below under point 7.

“4. The viable alternate hypotheses functionally are that (a) as suggested in ref 15, WT peptide is never presented effectively on the cell surface and that there are potentially TCRs that interact both with WT unflipped p6W and the p2L variant in both flipped and unflipped p6W states, with cross-reactivity possible particularly for beta-dominant TCRs, or (b) WT peptide is presented effectively as posited in this manuscript and all WT-specific TCRs are negatively selected during thymocyte development, leaving only TCRs that interact with the flipped p6W and thus only recognizing mutant pMHC. Is (was) it possible to isolate a WT-specific TCR from healthy volunteers? Was such an effort made? Is it likely that TCRs could recognize the unflipped neoantigen-MHC? Can this information be readily gleaned from MD? Is there any further evidence that the recognition mode of TCR3 and TCR4 is typical of TCRs found in humans that recognize this neoepitope? Could this be commented on, if so.”

This point relates to TCR recognition *in vitro* vs. *in vivo*, addressed above in response to Reviewer 2. We emphasize that we are not claiming that every PI3K α neoAg-specific TCR will behave as TCR4 and TCR3 here do – while it is intriguing that these two unrelated TCRs with different sequences trigger the same conformational change in the peptide, others may not – we know from other work that different TCRs can find different binding solutions on the same ligand (the structures of the DMF4 and DMF5 TCRs that bind MART-1/HLA-A2 is a good instructional example; see Borbulevych et al., JI 2011, PMID 21795600). In response to this part of the comment, we reinforced our statements that we are studying select TCRs, as highlighted on **page 18** in the Discussion.

We do not posit that the peptide is presented *in vivo* and drives negative selection. We have not yet found TCRs that recognize the WT peptide (although there are hints with some at very high peptide concentrations), and while this *could* be due to negative selection eliminating WT sensitive TCRs, we also know that the peptide binds poorly to HLA-A3 as the reviewer emphasizes.

We return to the main point of the manuscript as emphasized in the Introduction and Discussion: our focus is on the concordance between the inability to detect TCR binding with two TCRs, the lack of an *in vitro* response, and how peptide motional properties contribute to these, with the story being an instructional example of how TCR binding specificity can emerge. In response to this part of the comment, we added text in the Discussion that TCRs capable of recognizing the WT peptide may exist (first paragraph on **page 18**).

Lastly, per the reviewer’s question, the MD simulations provide some clues as to why recognition of the WT complex might be more difficult to find, and we allude to this on **pages 17-18** of the Discussion (beginning at the bottom of **page 17**), where we highlight how the simulations of the WT peptide show a

tendency for the tryptophan to stick out of the groove. We make note on **page 18** (end of first paragraph) that some TCRs could possibly bind in ways that are not perturbed by this dynamic and make a possible connection to force-dependent responses (see also responses to points 5 and 7 below).

“5. In the discussion about “structurally silent” recognition elements, the authors reference the A6 TCR system recognizing a cognate HTLVI Tax peptide and its closely related variants which yielded highly disparate functional T-cell activation results. While this seems germane to the present manuscript, as referenced, it is also appropriate to reference and discuss a recently published paper demonstrating dynamic differences between the TCR-pMHC in this system as studied with and without force (Chang-Gonzalez et al. eLife 2024; 13:e91881. DOI: <https://doi.org/10.7554/eLife.91881>). Relevantly, the application of force is also a variable not accounted for in static structural models on which Baker was a coauthor years ago. The eLife MD study with and without force application not only demonstrates dynamic allostery in the TCR system but offers the rational basis for explaining functional differences in structurally highly similar pMHC ligands interacting with the same TCR. Hence the omission of this reference and discussion in the current context is glaring.”

This is a good point, and we certainly agree relevant to TCR specificity. We recently discussed how force-dependent changes could influence TCR recognition (Ayres et al. JI 2023). Given that this influence arises from how structures move and evolve under force, it is certainly possible that peptide dependent tuning of peptide/MHC landscapes could alter force induced dynamic responses.

We note that in the referenced case, binding to the agonist and antagonist ligands are quantifiable, there are structures for both, and there are data indicating that some kind of signal is sent with both. For the WT peptide here we have none of that, precluding a discussion about more complex contributions to specificity. Given the relevance of the issue though, we added a paragraph to the Discussion (**page 19**, second to the last in the Discussion) regarding the impact of force on TCR specificity and cited the requested paper (new ref. 77) and others, keeping the comments more general as is appropriate.

We are particularly intrigued by the possibility that peptide induced alterations of pMHC dynamics could change dynamic responses under force and introduce that as a future direction. We make a brief note on this on **page 18** about other TCRs possibly achieving selectivity through this mechanism.

“6. The historical term “structurally silent” is, in fact, a misnomer and should be explicitly called out in the abstract, introduction and/or discussion. As this manuscript aptly shows, the differences are not silent except by virtue of the technique used by investigators to interrogate the structure, in this case being restricted to X-ray crystallography.”

We removed this term from the manuscript.

“7. Molecular dynamic considerations: Using unbiased molecular dynamics (MD) simulation as well as biased simulations, they argue that the motion of H2 in the WT peptide results in a high barrier for turning W6 towards the alpha2 helix of MHC. While the simulation and analyses performed are overall well-done, there are some concerns as follows:

The alpha2-facing orientation of W6 was observed in x-ray structures of the system with a bound TCR (PDB 7L1D & 7RRG), whereas all the simulations in this manuscript were performed in the absence of TCR. Without considering TCR, the of important the proposed mechanism is uncertain. Pointedly, the interaction with the TCR may well change the energy landscape associated with flipping of W6, and the low affinity for TCR of the WT system could be due to a different mechanism, e.g., simply due to a higher mobility of the N-terminal part of the WT peptide (Fig 4A) that cannot stabilize the bound state, and/or the less bulky WT H2 (compared to L2 of the mutant) makes the peptide slightly closer to the MHC floor (as they observed), in which case the motion of H2 per se may not be as important. Ignoring the interaction with TCR is the biggest shortcoming of this current MD study. This should be rectified.”

We fully agree that TCR binding changes the energy landscape describing the tryptophan conformational change and argue that the data support this, as we describe on **page 18** of the Discussion.

Indeed, traditional interpretations of structures would lead to this conclusion: in the TCR-free state the tryptophan is not flipped, but in the TCR-bound state it is. However, there are of course decades of structures being over-interpreted. This was the basis for our use of the Bta analogue of tryptophan (**Figs. 3D-G, page 7**), which strongly supports the structural observations and our interpretation of the importance of the flip of the tryptophan.

In further response to this comment, notably our lack of direct study of a TCR complex, we collected additional NMR data on a complex between TCR3 and the neoAg/HLA-A3 ligand (**new Fig. 8, described on pages 15-16 of the Results**). The results are fully consistent with the flip of the tryptophan being required for TCR binding, as unlike in the free pMHC, we only see a single peak describing the large complex. To help interpret this data, we also performed MD simulations on the complexes with TCR3 and TCR4, looking at the motional properties of the tryptophan (**new Fig. 8, discussed on page 16**). Consistent with the new NMR data and the crystallographic structures, pTrp6 is more restrained in the complex than in the free MHC protein. These same simulations show Leu2 of the neoantigen remains rigid in the complex (**new Supplemental Fig. 4**).

We also performed MD on hypothetical TCR complexes in which the TCR is not flipped – there is no evidence that these complexes exist to any appreciable extent (as shown by X-ray, NMR, and the Bta experiment), but we thought it would be an interesting exercise. In these simulations the pTrp6 side chain remains highly mobile, inconsistent with the NMR data. As this is purely speculative, we decided to leave this data out of the paper, but the data are shown below (compare with **Fig. 8** in the revised manuscript, which describes MD starting with the crystallographic structures).

Lastly, we do not observe TCR binding shifting the peptide closer to the floor of the binding groove (other than what occurs with Trp6). Below is a figure showing the neoAg bound to TCR3 and TCR4, along with both structures of the free neoAg. The superimposition is through residues 1-180 of the HLA-A3 peptide binding groove. There is no apparent movement of the peptide closer to the floor, unless one considers the backbone at positions 5 and 6 where the flip of the tryptophan occurs. There is a very small 0.5 Å movement of the C α of His3 seen with TCR4 (shown by an asterisk) with one of the two free neoAg structures when looking at the binding of TCR4, but this is absent when comparing the other free neoAg structure and pales in comparison to the dramatic rotation of Trp6.

“Could the interaction between TCR and pMHC differentially augment W6 flipping or otherwise alter recognition of WT vs mut pMHC with or without force?”

This is possible and discussed at a high level in a new paragraph in the Discussion (**page 19**). We also make a brief note on this on **page 18** (end of first paragraph) about alternate TCRs achieving selectivity. As noted above though, for this particularly case, there is no indication that the WT complex is recognized biochemically or functionally, so anything we might say about this particular system is purely speculative. We thus restrict our comments to the more general as is appropriate.

*“Does this relate to differential recognition in the context of highly related HLA molecules (HLA-A*03:01 serves to mediate mutant T-cell recognition whereas HLA -A*03:02 and -A* 11:01 do not) noted in their prior Nature Medicine 2022 study?”*

This is a good and prescient question. We have not studied HLA-A11, but we have studied HLA-A*0302 with crystallography, NMR, and simulations. A*0302 differs from A*0301 at two positions, L156-*Q and E152-*V at the α 2 helix. These positions are very close to Trp6 in the unbound peptide/HLA-A3 complex.

Without putting the whole story here (and it is still incomplete), the data indicate that in A*0302 the two mutations alter the motional properties of the peptide and thus the propensity for the tryptophan to flip. We are working this up for a follow-up manuscript, but a key piece of the data is below, where we show ¹⁹F NMR data on the neoAg bound to A*0302 – there is a single peak, unlike what we see with A*0301 (compare with **Fig. 7** in the revised manuscript, which shows multiple peaks). The crystal structure of the neoAg bound to A*0302 is again essentially the same as with A*0301, as also shown below. Our current hypothesis is that the peptide is unable to flip in A*0302, preventing recognition by the TCRs we studied.

(Redacted)

“How do the authors think about patients whose T cell recognize the neoepitope, yet CTL effector function may not be adequate to mediate tumor protection since a clinical tumor is evident? Is efficiency inadequate because of requisite structural rearrangements? This last question is not meant to be answered but rather food for thought.”

This is a good question and one related to efficacy in general. We might hypothesize that a more potent TCR might be needed or perhaps that the TCR is too potent, leading to exhaustion. We note that the TCRs we identified in the 2022 study and studied here were from healthy donors.

“Based on difference in the motion of the residue at position 2 affecting the behavior of W6 along the peptide, the authors use the term ‘dynamic allostery’. It is unclear whether use of the term is suitable here. The WT peptide has H2 and the mutant has L2, with the latter peptide residue being bulkier and more hydrophobic. As mentioned above, is it the motion of H2 that prevents W6 flipping, or the conformation of the peptide that differ slightly from L2 that is responsible for the difference? How are these motions impacted by TCR ligation?”

The term allostery is phenomenological and originated with enzymology where it refers to the binding of allosteric effectors that modulate the binding of a substrate. Over time, this has evolved to reflect “action at a distance” in a protein, frequently involving fast or slow protein dynamics and manifesting via changes in coupled fluctuations due to binding of another molecule, a substitution, mutation, etc. Allostery is just as frequently described as being across domains as it is at closely connected sites within proteins. The latter is apt here, and our data – particularly the NMR data on the free peptide/HLA-A3 complexes – show unambiguously that Trp6 moves differently depending on the identity of the position 2 anchor. Our use of allostery is thus appropriate.

The second question is more semantic – regardless of the reason for the flip of Trp6 in the neoAg (or the lack of the flip in the WT peptide), all experimental data indicate that the flip is necessary for the binding of TCR3/4. Our MD data indicates that this can be attributable to knock-on (i.e., allosteric) dynamic effects resulting from anchor modification. How other subtle differences in peptide conformation contribute to the specificity of the TCR we don’t know, but we do know that the new neoAg/HLA-A3 complex is even *more* similar to the WT peptide/HLA-A3 complex (**updated Fig. 1A**). The data all point to the impact of anchor modification on protein dynamics, very similar to what has been seen (with least deconstruction) in other systems such as recognition of anchor modified gp100 peptides (ref. 6 in the revised manuscript).

As discussed above in response to point 7, we included an analysis of the motions of the tryptophan when ligated by a TCR, both by NMR and simulation – Trp6 of the peptide is rigidified as expected. We also included an analysis of Leu2, whose motions are unchanged upon TCR binding (**new Fig. 8B** and **Supplemental Fig. 4**).

Minor:

“1. Please provide reference for CD107a (LAMP-1) assay in the methods section. Why this vs. IL2 or CD69? Have the authors found this technique more sensitive or otherwise advantageous over the others or just comparable as a point of information.”

CD107a is an acute marker of activation-induced degranulation and correlates with CD8 T cell cytolytic function (PMID 14580882). Our previous work reported on the polyfunctionality of the PI3K α neoAg TCRs (ref. 15) - Supplementary Fig. 5a from that work is reproduced below and demonstrates that CD107a levels are comparable to multiple cytokines that display similar kinetics of upregulation at 46h post-stimulation (unlike CD69 which achieves peak upregulation at 24h post-stimulation). As requested, the above listed citation is now cited (ref. 101 in the revised manuscript).

"2. On p6 para 4 , "...feature of PIK3CA neoantigen recognition by multiple TCRs, ..." , should the word "multiple" be changed to "two" as there are only two defined structures?"

Correct – the change has been made, and this better reflects our language on the previous page.

"3. Is there a volumetric measurement available for the 3D conformational sampling represented by occupied voxels in Figs 4d-e and Fig 5a? It might be helpful to the reader to have some quantitation?"

This is a good suggestion, and the values have been added to the relevant figures.

"4. Regarding the ED Fig 6e -124.92 ppm "degradation" peak in the 19F spectra, is it likely that this is free peptide non-specifically associated with unfolded HLA?"

This is our current interpretation, as the peak is too broad to be free peptide as shown in **Figs 7A and 7E**. This is noted on **page 15** in the Results.

"5. Please provide names of Bruker pulse sequences used for CEST and EXSY with references. Please provide SW, number of increments and scans for the CEST. Also, were spectra processed with Topspin? Was the unsaturated reference with no pulse or off-resonance pulse?"

Changed as requested – with the inclusion of the NMR on the complex, the NMR Methods section has been rewritten, with direct references to pulse program names, spectral widths, increments, number of transients, as well as acquisition and processing details. Citations to applications of CEST and EXSY for fluorine detection in proteins have also been added.

"6. Histidine protonation states chosen for simulation are not mentioned in the manuscript. This is especially important for H2 and H3 of the peptide, which could impact dynamics of the peptide."

The simulations for the WT peptide were performed with the His2 deprotonated, as indicated by its pKa of 4.9 (described on **page 10, first paragraph**). This is now explicitly mentioned in the methods section (**pages 21 and 23**).

The computed pKa of the more exposed His3 ranged from 5.8 for the WT peptide to 6.6 for the neoantigen. At pH 7.4, these translate into ~2% charged for WT and ~14% charged for the neoantigen. To evaluate the potential impact of the small amount of neoantigen that would be protonated and charged at His3, we performed additional peptide/MHC MD simulations with pHis3 charged. The results were unchanged. This is now included in **a new Supplemental Fig. 5B** and discussed on **pages 23 and 24**).

"7. p9 "rotation of Tyr99 was slightly (~1ns) preceded by rotation of pHis2": Without more quantitative analysis, this statement cannot be concluded from fig 5C"

We removed this component as part of the effort to simplify some of the molecular dynamics components as requested by Reviewer 1.

Response to reviewers

We thank the editor, editorial staff, and the reviewers for their time and effort. No changes were requested from the reviewers.

Changes were made based on the editor's request as documented in the focused editorial checklist.